# Kinase-dead ATR differs from ATR loss by limiting the dynamic exchange of ATR and RPA

Demis Menolfi[1], Wenxia Jiang[1], Brian J. Lee[1], Tatiana Moiseeva [2], Zhengping Shao[1], Verna Estes[1], Mark G. Frattini[4], Christopher J. Bakkenist[2,3] & Shan Zha[1,5]

ATR kinase is activated by RPA-coated single-stranded DNA (ssDNA) to orchestrate DNA damage responses. Here we show that ATR inhibition differs from ATR loss. Mouse model expressing kinase-dead ATR ($Atr^{+/KD}$), but not loss of ATR ($Atr^{+/-}$), displays ssDNA-dependent defects at the non-homologous region of X-Y chromosomes during male meiosis leading to sterility, and at telomeres, rDNA, and fragile sites during mitosis leading to lymphocytopenia. Mechanistically, we find that ATR kinase activity is necessary for the rapid exchange of ATR at DNA-damage-sites, which in turn promotes CHK1-phosphorylation. ATR-KD, but not loss of ATR, traps a subset of ATR and RPA on chromatin, where RPA is hyper-phosphorylated by ATM/DNA-PKcs and prevents downstream repair. Consequently, $Atr^{+/KD}$ cells have shorter inter-origin distances and are vulnerable to induced fork collapses, genome instability and mitotic catastrophe. These results reveal mechanistic differences between ATR inhibition and ATR loss, with implications for ATR signaling and cancer therapy.

[1] Institute for Cancer Genetics, Department of Pathology and Cell Biology, College of Physicians & Surgeons, Columbia University, New York, NY 10032-3802, USA. [2] Department of Radiation Oncology, University of Pittsburgh School of Medicine, Hillman Cancer Center, Research Pavilion, Suite 2.6, 5117 Centre Avenue, Pittsburgh, PA 15213-1863, USA. [3] Department of Pharmacology and Chemical Biology, University of Pittsburgh School of Medicine, Hillman Cancer Center, Research Pavilion, Suite 2.6, 5117 Centre Avenue, Pittsburgh, PA 15213-1863, USA. [4] Department of Medicine, College of Physicians & Surgeons, Columbia University, New York, NY 10032-3802, USA. [5] Division of Pediatric Oncology, Hematology and Stem Cell Transplantation, Department of Pediatrics, College of Physicians & Surgeons, Columbia University, New York, NY 10032-3802, USA. Correspondence and requests for materials should be addressed to S.Z. (email: sz2296@columbia.edu)

A TR kinase belongs to the phosphoinositide (PI) 3-kinase-related protein kinases (PI3KKs) family that also includes ATM and DNA-PKcs. In contrast to ATM and DNA-PKcs that are primarily activated by DNA double strand breaks (DSBs), ATR is recruited to and activated by RPA-coated ssDNA filaments through interaction with its obligatory partner ATRIP[1,2]. In addition to resected DSBs, ssDNA/RPA filaments can also be generated on the lagging strand during DNA replication, on R-loops during transcription, on the non-homologous regions of the X and Y chromosomes during meiosis, and other processes, thus giving ATR the unique ability to respond to a broad range of DNA structures[3]. Once activated, ATR phosphorylates numerous substrates, especially its effector kinase CHK1, and together ATR and CHK1 activate the intra-S and G2/M checkpoints, suppress origin firing, stabilize stalled replication forks, prevent premature mitosis, and eventually promote fork restart[3]. Given their critical role in DNA replication, complete loss of ATR or CHK1 is incompatible with normal embryonic development or sustained proliferation of cells in culture[4–6]. Therefore it is unexpected that specific and highly potent ATR kinase inhibitors are very well tolerated in preclinical animal models and clinical trials[7] and display synergistic effect with cisplatin and other genotoxic chemotherapies, suggesting that ATR inhibition might differ from ATR deletion.

While ATR is recruited and activated by RPA-coated ssDNA, full ATR activation also requires additional factors[8], including RAD17, RAD9-RAD1-HUS1 (9-1-1), and the allosteric activators TOPBP1 or ETAA1[9–13], all of which are associated with chromatin at the time of ATR activation. Indeed, ATR forms stable foci (>30 min) at the DNA damage sites and the phosphorylated forms of several ATR substrates, including RAD17, CHK1, RPA, and ATR itself, are also enriched in the chromatin fraction[14,15]. Based on these and other findings, it was proposed that the active ATR remains tethered to the sensor-DNA complex at the chromatin, where it phosphorylates its substrates. The model makes two predictions. First, ATR substrates have to be able to cycle through the active ATR to get phosphorylated. Second, the RPA-coated ssDNA can only activate one round of ATR. However, a large number of substrates for ATR and its yeast ortholog Mec1 have been identified from proteomic studies[16,17]. Not all of them show evidence for looping through the DNA lesion. For example, during male meiosis, ATR phosphorylates histone H2AX molecules embedded in chromatin loops kilobases away from the initiating DNA lesion[18]. Moreover, heterozygous $Atr^{+/−}$ mice, though viable, display tumor susceptibility and dose-dependent reduction of CHK1 phosphorylation, suggesting that under physiological stress, more than half of the cellular ATR pool may be activated simultaneously[19].

On this note, both ATM and DNA-PKcs are transiently recruited to the site of DNA damage via their specific sensor complex (e.g., MRN for ATM, and Ku70-Ku80 for DNA-PKcs). Despite normal development of ATM- and DNA-PKcs-null mice[20–23], we and others previously showed that mouse models expressing kinase-dead (KD) ATM ($Atm^{KD/KD}$) or DNA-PKcs ($DNA\text{-}PKcs^{KD/KD}$) exhibit severe genome instability and die during embryonic development[24–26]. Loss of Ku70-Ku80 prevents the recruitment of DNA-PKcs-KD and rescues the embryonic lethality of $DNA\text{-}PKcs^{KD/KD}$ mice, suggesting that catalytically-inactive DNA-PKcs physically blocks the repair of DSB ends[26]. Similar observations were also made for ATM-KD[27]. Thus, the question is whether ATR, like ATM and DNA-PKcs, has a kinase-dependent structural function during DNA repair, which will explain the difference between ATR inhibition vs ATR loss.

Here, we present the first knock-in mouse model expressing kinase-dead (KD) ATR protein ($Atr^{+/KD}$). We report that $Atr^{+/KD}$ mice display ssDNA toxicity at the non-homologous regions of the X–Y chromosomes during meiosis and at telomeric and rDNA loci during mitosis, which lead to male sterility and lymphocytopenia, respectively. Using live cell imaging, we found that the apparent stable ATR foci at the DNA damage site reflect the rapid exchange of active ATR. And importantly, ATR kinase activity is necessary for ATR exchange, which in turn promotes robust DNA damage responses. Moreover ATR-KD, but not the loss of ATR, traps a subset of RPA on chromatin, where RPA is eventually hyper-phosphorylated by ATM and DNA-PKcs, and together with stalled ATR-KD, compromises subsequent repair (e.g., RAD51 foci formation). Thus, our findings uncover the kinase-dependent exchange of ATR at the DNA damage sites, which explains the ssDNA-dependent effects of ATR-KD protein during physiological and damage-induced processes, revealing one molecular mechanism that explains the difference between ATR kinase inhibition and ATR loss.

## Results

**$Atr^{+/KD}$ male mice have spermatogenesis defects**. We generated a mouse model expressing ATR-KD protein (D2466A) from the endogenous $Atr$ locus ($Atr^{+/KD}$) (Fig. 1a and Supplementary Fig. 1a)[28,29]. We chose this mutation, since this is a highly conserved catalytic residue shared by ATR from all eukaryotic origins (D2224 in yeast Mec1 and D2475 in human ATR) and by the related ATM kinase and DNA-PKcs[24,26]. A structural study mapped the D2224 on the catalytic loop of Mec1, essential for ATP transfer[29]. Importantly, the corresponding mutation in yeast (D2224A)[30] and human (D2475A)[31] orthologues of ATR eliminates ATR kinase activity without affecting ATR protein stability. Indeed, ATR-KD transcript and protein are stable and ATR-KD protein lacks measurable kinase activity (Supplementary Fig. 1b and d). While $Atr^{+/KD}$ mice were born at the expected Mendelian ratio and of normal size (Supplementary Fig. 1e, f), $Atr^{KD/−}$ mice could not be found at birth (Supplementary Fig. 1g), consistent with the essential role of ATR kinase activity in embryonic development. Notably, while female $Atr^{+/KD}$ mice are fertile and have normal litter size in comparison to $Atr^{+/−}$ females (Supplementary Fig. 1h), male $Atr^{+/KD}$ mice are sterile with a 100-fold reduction of sperm count and markedly reduced testis weight (Fig. 1b, c). This was unexpected, since male $Atr^{+/−}$ mice are fertile and have normal sperm counts and testis weight (Fig. 1b, c)[32]. Histologic analyses of $Atr^{+/KD}$ mice revealed a major reduction of mature spermatids in both the epididymis and testes and a concurrent accumulation of large primary spermatocytes in the seminiferous tubules (Fig. 1d), consistent with a developmental arrest at primary spermatocytes. TUNEL staining uncovered a significant number of seminiferous tubules from $Atr^{+/KD}$ mice that have numerous ($n > 3$) apoptotic cells with morphological features that are consistent with meiotic prophase I cell death (Fig. 1d and Supplementary Figure 2a). By 23 weeks, when the testes from $Atr^{+/+}$ mice remain packed, the testes from $Atr^{+/KD}$ mice are often occupied by enlarged cysts (Supplementary Fig. 2b). To determine at which stage within the meiotic prophase I the spermatogenesis is blocked, we analyzed meiosis progression from 3 weeks old mice, when the first wave of spermatogenesis occurs. The result uncovered a significant reduction of diplotene cells specifically in $Atr^{+/KD}$ mice, suggesting blockade at pachytene to diplotene transition (Fig. 1e and Supplementary Fig. 2c). At the pachytene stage, ATR kinase is recruited to and activated by the ssDNA accumulated at the non-homologous region of the X–Y chromosomes (X–Y body)[33]. Activated ATR phosphorylates

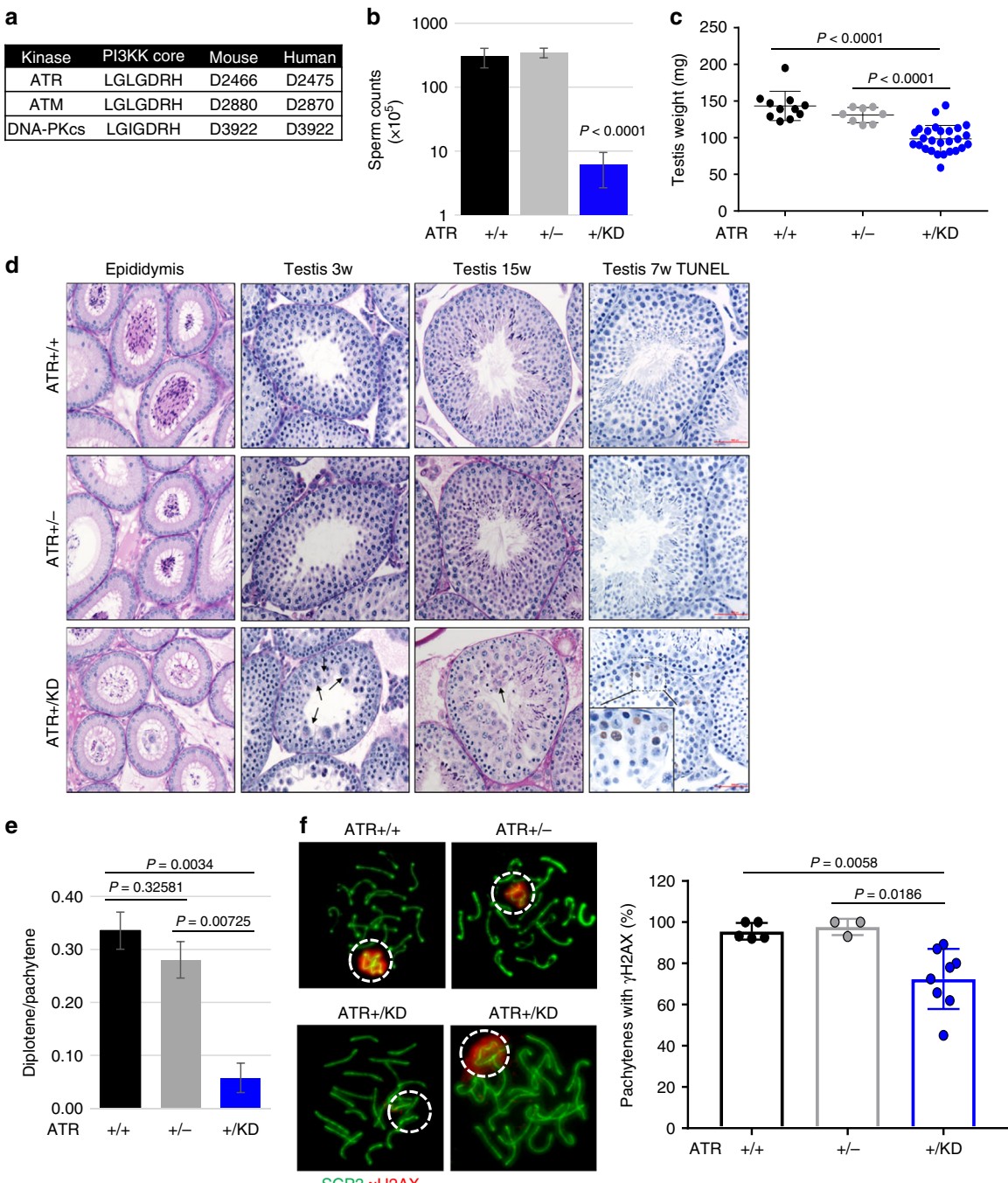

**Fig. 1** $Atr^{+/KD}$ male mice display defects in spermatogenesis and meiotic sex chromosome inactivation. **a** Conservation and relative amino acid number of aspartate (D) residue in the PI3KK core of ATR, ATM, and DNA-PKcs in mouse and human. **b** The histogram reports the means ± SD of the sperm counts derived from 6 $Atr^{+/+}$, 5 $Atr^{+/-}$, and 14 $Atr^{+/KD}$ male mice. Unpaired two-tailed $t$ test was used. **c** Testes from several 12–37 weeks old $Atr^{+/+}$, $Atr^{+/-}$, and $Atr^{+/KD}$ mice were weighed. Mann–Whitney test was used. **d** Representative histological sections of H&E staining of epididymis and testis (3 weeks and 15 weeks) of $Atr^{+/+}$, $Atr^{+/-}$, and $Atr^{+/KD}$ mice. TUNEL assay was performed on 7 weeks old mice and representative images are reported. **e** The means ± SD of the ratio between diplotene and pachytene cells for $Atr^{+/+}$, $Atr^{+/-}$, and $Atr^{+/KD}$ mice are reported and unpaired two-tailed $t$ test was used for the statistical analysis. **f** Representative SCP3 and γH2AX immunofluorescence images of spermatocytes spreads are shown for $Atr^{+/+}$, $Atr^{+/-}$, and $Atr^{+/KD}$ mice. The distribution of the mean percentages of positive γH2AX pachytenes is reported. A total of 587, 214, and 647 pachytenes were analyzed from 5 $Atr^{+/+}$, 3 $Atr^{+/-}$, and 8 $Atr^{+/KD}$ mice, respectively. Unpaired two-tailed $t$ test was used for the statistical analysis. Source data are provided as a Source Data file

H2AX to form γ-H2AX on the chromosome loops, which is necessary for transcriptional silencing of X–Y associated genes (termed meiotic sex chromosome inactivation; MSCI) and further spermatogenesis[18,34]. While almost all pachytene spermatocytes from age-matched $Atr^{+/+}$ and $Atr^{+/-}$ mice are positive for γ-H2AX, up to 50% of pachytene spermatocytes from $Atr^{+/KD}$ mice display little or no γ-H2AX staining (Fig. 1f). Correspondingly, the expression of X and Y chromosome-associated genes increases ~2 folds in the testes of 7-week-old $Atr^{+/KD}$ mice, indicating MSCI defects (Supplementary Fig. 2d). Meanwhile,

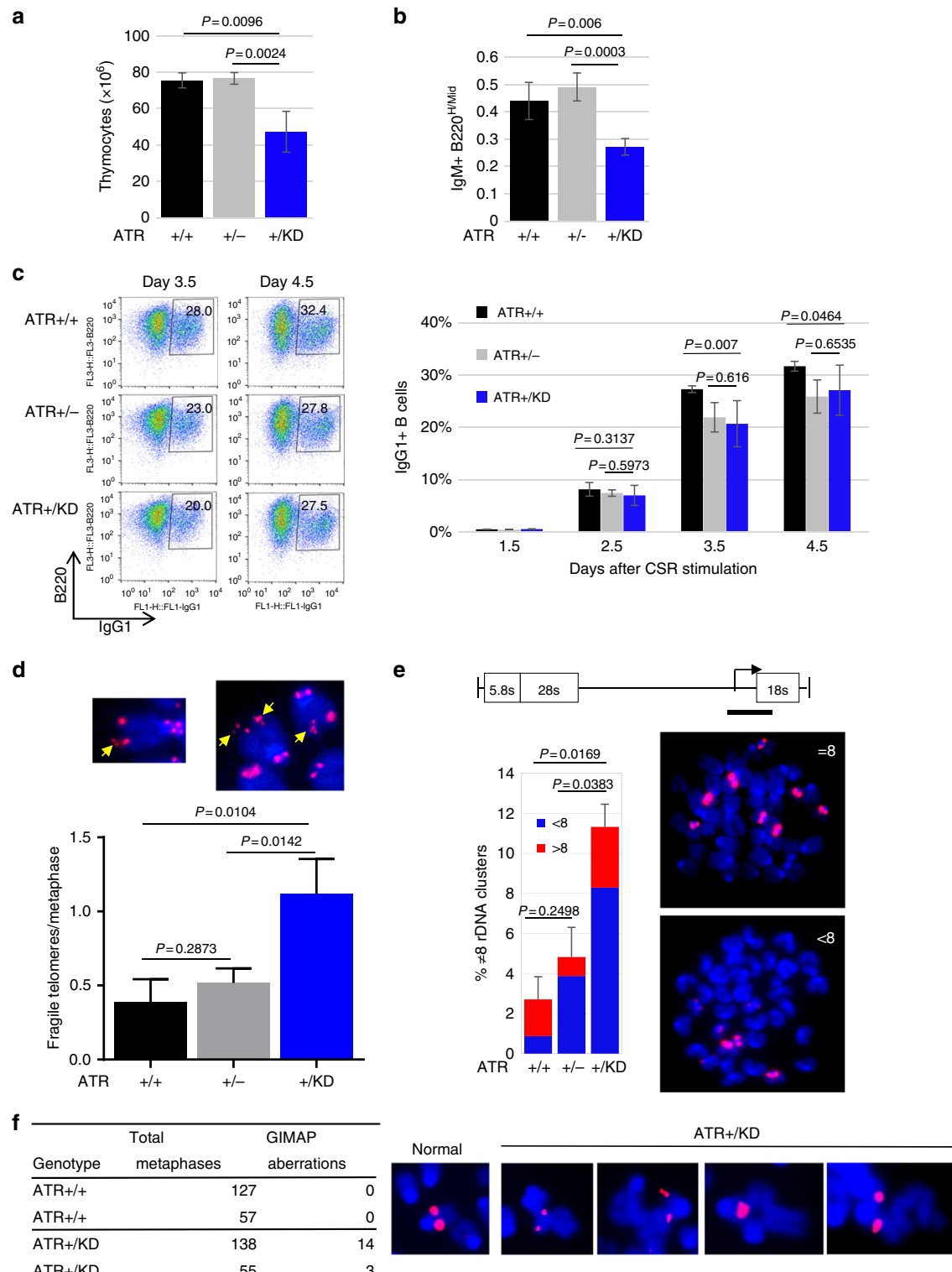

the appearance and frequency of RAD51 foci, a marker for meiosis recombination (Supplementary Fig. 2f), and MLH1 foci, a marker for meiosis cross over (Supplementary Fig. 2e), were not significantly altered in the $Atr^{+/KD}$ spermatocytes. Together with the normal fertility of female $Atr^{+/KD}$ mice, our data suggest that ATR-KD does not block meiosis recombination at autosomes, but displays selective dominant-negative effects at the X–Y body, where ssDNA accumulates at the non-homologous region during meiotic recombination.

**Mild lymphocytopenia in $Atr^{+/KD}$ mice**. To ascertain whether ATR-KD has also ssDNA-dependent dominant-negative effects during mitosis, we analyzed lymphocyte development. Thymus cellularity and thymus net weight decreased nearly ~50% in $Atr^{+/KD}$ mice relative to $Atr^{+/+}$ and $Atr^{+/-}$ mice (Fig. 2a and Supplementary Fig. 3c). Yet, the relative frequency of different blood cell types (e.g., lymphoid, myeloid, or erythroid) or between different developmental stages (i.e., progenitors vs native lymphocytes) were quite similar among the $Atr^{+/KD}$, $Atr^{+/-}$, and

**Fig. 2** Mild lymphocytopenia in $Atr^{+/KD}$ mice and genomic instabilities at telomeres, rDNA repeats, and fragile sites in $Atr^{+/KD}$ cells. **a** Thymocytes were counted with Trypan Blue from several $Atr^{+/+}$, $Atr^{+/-}$, and $Atr^{+/KD}$ thymuses. The means ± SD are reported and unpaired two-tailed $t$ test was used. **b** The means ± SD of the ratio of IgM+B220$^{high}$ re-circulated B cells vs IgM+B220$^{mid}$ naive B cells in the bone marrow of $Atr^{+/+}$, $Atr^{+/-}$, and $Atr^{+/KD}$ mice are shown. Unpaired two-tailed $t$ test was used. **c** Purified naive CD43- B cells from spleens of $Atr^{+/+}$, $Atr^{+/-}$, and $Atr^{+/KD}$ mice were stimulated to CSR in vitro. Representative flow cytometry profiles with IgG1 and B220 staining are reported at days 3.5 and 4.5. The mean percentages ± SD of IgG1+ cells are plotted. Unpaired two-tailed $t$ test was used. $P$ values between $Atr^{+/+}$ and $Atr^{+/-}$ are 0.4592 (day 2.5), 0.0741 (day 3.5), and 0.0768 (day 4.5). $P$ values between $Atr^{+/KD}$ and controls are shown above the bar graph. **d** The means ± SD of fragile telomeres per metaphase were plotted for $Atr^{+/+}$, $Atr^{+/-}$, and $Atr^{+/KD}$ B cells from three independent experiments and unpaired two-tailed $t$ test was used. Representative images of fragile telomeres, indicated by the yellow arrows, are shown. **e** Metaphase spreads from $Atr^{+/+}$, $Atr^{+/-}$, and $Atr^{+/KD}$ B cells were stained with DAPI and a rDNA probe by FISH (two independent experiments). The bars show the mean percentages ± SD of metaphases that contain a number of rDNA clusters that is different from 8. Unpaired two-tailed $t$ test was used. Representative metaphases with 8 or ≠8 clusters are shown. **f** FISH on GIMAP locus was carried out on metaphase spreads from $Atr^{+/+}$ and $Atr^{+/KD}$ B cells treated with 0.25 mM HU. Two independent experiments were performed and the number of metaphases analyzed by FISH and the number of chromosome aberrations at the GIMAP locus are reported in the table. An example of a normal GIMAP locus and few examples of breaks and fusions near GIMAP locus found in $Atr^{+/KD}$ B cells are shown. Source data are provided as a Source Data file

$Atr^{+/+}$ mice analyzed (Supplementary Fig. 3a, b). The relative frequency of antigen experienced IgM+B220$^{high}$ re-circulated B cells over naive IgM+B220$^{Mid}$ cells is significantly lower in the bone marrow of $Atr^{+/KD}$ mice, implying potential defects in B cell maturation (Fig. 2b and Supplementary Fig. 3a, and b). Thus, we isolated naive $Atr^{+/KD}$ and control B cells to test their ability to respond to activation. Cytokines (anti-CD40 and IL4) activated $Atr^{+/KD}$ splenic B cells initiated class switch recombination (CSR) efficiently, but the frequency of switched, thus IgG1+, $Atr^{+/KD}$ B cells was significantly lower than that of $Atr^{+/+}$ B cells at both day 3 and day 4 (Fig. 2c). $Atr^{+/-}$ B cells also showed a trend toward reduced CSR at days 3 and 4, but the data did not reach statistical significance when compared to either $Atr^{+/+}$ or $Atr^{+/KD}$ B cells (Fig. 2c). Given the lack of stage-specific block in $Atr^{+/KD}$ lymphocytes and the need for continuous cell proliferation during both thymocyte development and B cells maturation (including but not limited to CSR), we reason that the lympho-cytopenia in $Atr^{+/KD}$ mice might reflect the accumulation of a mildly impaired mitosis, and perhaps genomic instability, during the numerous rounds of proliferation in vivo.

**Telomere and rDNA genomic instabilities in $Atr^{+/KD}$ B cells.** Next, we examined genomic instability at telomeres, rDNA repeats, and fragile sites, where ssDNA might accumulate during DNA replication[35–38]. In cytokine activated $Atr^{+/KD}$ B cells, cytogenetic analyses with telomere and rDNA specific probes revealed increased telomere fragility (defined by >1 telomere focus per chromatid)[35] (Fig. 2d) and ribosomal DNA instability, as measured by the fraction of metaphases that possess aberrant numbers of rDNA clusters (i.e., greater or less than 8 clusters) (Fig. 2e). The number and location of rDNA clusters vary among different inbred laboratory strains of mice and the 129 strain has 8 rDNA clusters detected by in situ hybridization[39]. Using a FISH probe covering part of the murine rDNA repeat (derived from BAC BK000964)[40] (see Methods for details), we detected 8 pairs of para-centrometic signals in more than 97% of metaphases derived from $Atr^{+/+}$ or $Atr^{+/-}$ cells (129 Sv background) (Fig. 2e). In contrast, nearly 8% metaphases from $Atr^{+/KD}$ cells (also in 129 Sv background) have <8 clusters and another 3% have >8 clusters, implying instability in the rDNA loci (e.g., aberrant recombination within or between rDNA loci leads to variable number) (Fig. 2e). We also found that the expression of B cell-specific early replicating fragile sites (ERFSs) that were initially defined in hydroxyurea (HU) treated activated B cells[41], is higher in HU-treated $Atr^{+/KD}$ B cells relative to HU-treated $Atr^{+/+}$ B cells (Supplementary Fig. 3d), promoting us to analyze cytogenetically defined breaks at GIMAP—one of the best characterized ERFSs in activated B cells. Using FISH probes (BAC, RP24-371A23 and RP24-387K23), we did not find any evidence for

breaks at or near the GIMAP locus in 184 metaphases from $Atr^{+/+}$ B cells, while identified 17 aberrations including breaks, radials, and fusions adjacent to GIMAP locus in 193 metaphases (8.8 per 100 metaphases) from $Atr^{+/KD}$ B cells (all treated with mild HU, 0.25 mM) (Fig. 2f). Together, these findings suggest that expression of the ATR-KD compromises the stability of difficult-to-replicate genomic regions (i.e., telomeres, rDNA, and fragile sites) in mitotic cells, where ssDNA might accumulate during replication[42].

**Kinase-dependent exchange of ATR on damaged chromatin.** ATR is recruited and activated on RPA-coated ssDNA. So we directly tested whether the phenotype of $Atr^{+/KD}$ mice and cells are due to alteration of ATR dynamics on ssDNA. To perform live cell imaging, we took advantage of previously characterized U2OS cells with stably integrated GFP-ATR[43]. Consistent with the previous reports[28], GFP-ATR was evenly dispersed throughout the nucleus before laser micro-irradiation (405 nm) (Fig. 3a). Upon micro-irradiation, GFP-ATR foci gradually appear and reach maximum fluorescence intensity at ~10 min. The intensity of ATR foci remain unchanged for at least 30 min (Fig. 3a and Supplementary Fig. 4a). This is in contrast to DNA-PKcs or ATM, which are immediately recruited to micro-irradiation induced foci and promptly disassociate within 30 min[44,45]. Transiently expressed GFP-RPA and GFP-ATRIP displayed similar kinetics (Supplementary Fig. 4a, b), suggesting that the initial delay might reflect the time needed for end-resection and generation of RPA-coated ssDNA. ATR kinase inhibitor did not alter the initial recruitment of GFP-ATR (Supplementary Fig. 4a). RFP-ATR-KD, transiently transfected in U2OS cells, can also be robustly recruited to the site of DNA damage (Supplementary Fig. 4b), indicating that ATR kinase activity is not required for the initial recruitment of ATR itself. To determine whether the stable GFP-ATR foci (10–30 min after initial damage) are made up of stationary ATR molecules or rapid exchange of many different ATR, we measured fluor-escence recovery after photobleaching (FRAP) with a 488 nm laser specific for GFP. Remarkably, immediately after photo-bleaching, the GFP-ATR intensity bounced back rapidly ($t_{1/2}$ = 13.36 ± 0.98 s) and nearly completely (maximal recovery = 88.5 ± 3.4%), indicating that almost all active ATR kinase exchanges rapidly on the RPA-coated ssDNA (Fig. 3b). Strikingly, ATR kinase inhibitor (VE-821, 10 μM) severely limited the recovery of ATR foci after photobleaching (maximal recovery fraction dropped from 88.5 ± 3.4% to only 39.4 ± 2.4%), while CHK1 inhibitor (LY2603618, 10 μM) caused at most a mild reduction in the maximal recovery fraction (77.2 ± 2.4%) (Fig. 3b). Nei-ther ATR nor CHK1 inhibition significantly affected the $t_{1/2}$, defined as the time needed to reach 50% of maximum recovery

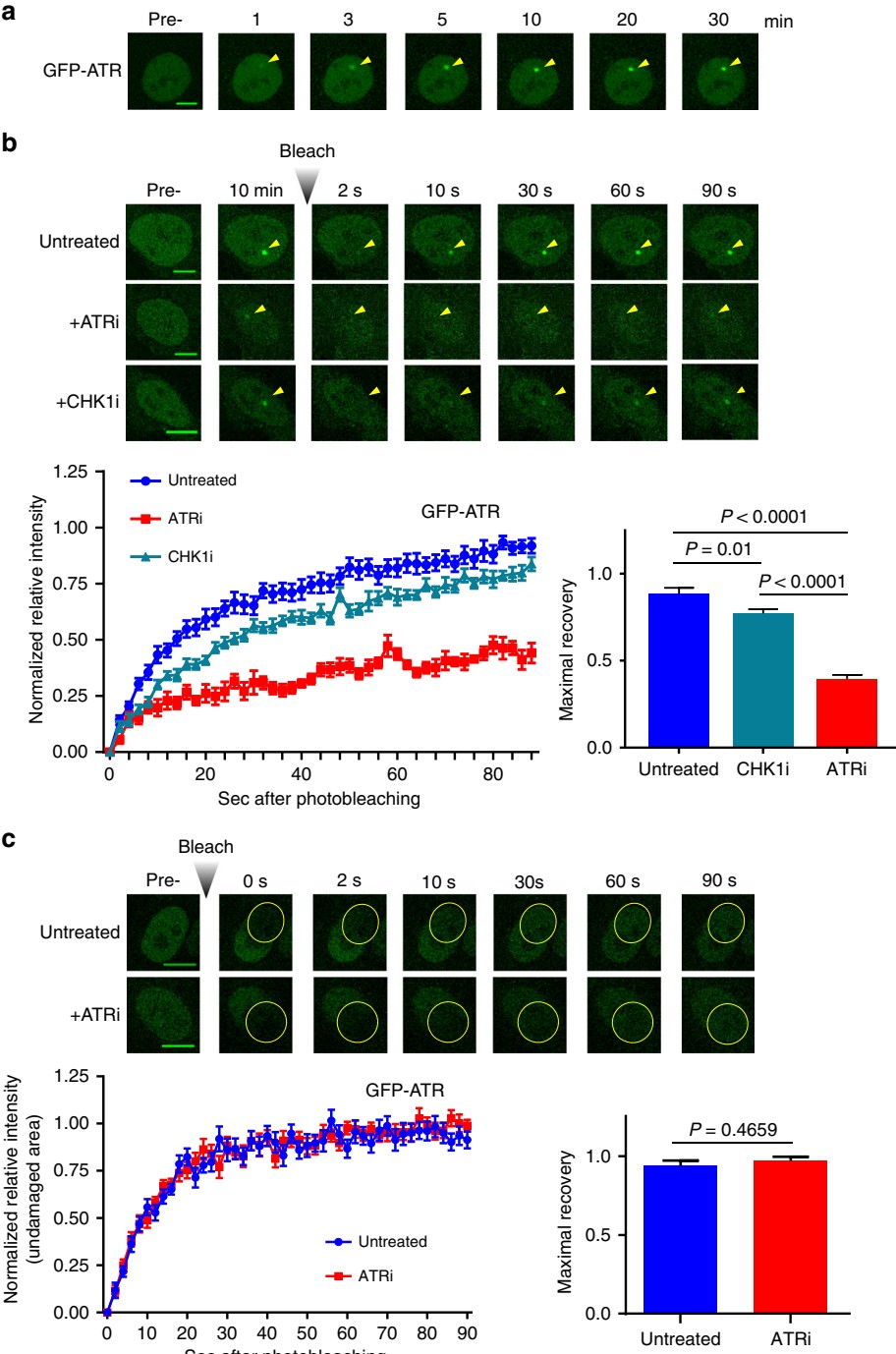

**Fig. 3** The rapid exchange of ATR on damaged DNA is dependent on its kinase activity. **a** The kinetics of GFP-ATR recruitment to sites of DNA damage. U2OS cells stably expressing GFP-ATR were monitored by confocal microscopy at the indicated time points after micro-irradiation (405 nm). Images of a representative cell are shown (scale bar is 10 μm). **b** FRAP analysis of GFP-ATR in U2OS cells after laser-induced foci formation, untreated or following treatment with ATRi (VE-821) or CHK1i (LY2603618) at 10 μM concentration. After photobleaching, recovery of the focus signal was monitored up to 90 s. The means ± SEM of the maximal recovery of GFP-ATR are reported for every condition analyzed and unpaired two-tailed *t* test was used for the statistical analysis. Representative cells are shown. Scale bar is 10 μM. **c** Quantification of the recovery of GFP-ATR relative intensity in an undamaged area after bleaching, in untreated or ATRi treated U2OS cells. The means ± SEM of the maximal recovery of GFP-ATR are reported and unpaired two-tailed *t* test was used for the statistical analysis. Representative cells are shown. Scale bar is 10 μM. Source data are provided as a Source Data file

(17.65 ± 2.75 and 19.32 ± 1.05 s, respectively). Importantly, the exchange of free ATR protein (i.e., not at a DNA damage site) was not affected by ATR inhibition (Fig. 3c). The maximal recovery after photobleaching for the transiently expressed RFP-ATR-KD was also lower than that of RFP-ATR-WT

(Supplementary Fig. 4c), suggesting the presence of endogenous WT-ATR is not sufficient to rescue the exchange defect in ATR-KD. Moreover, transient expression of RFP-ATR-KD reduced the exchange of stable expressed GFP-ATR-WT in the same cell, as such the recovery rate of RFP-ATR-KD and GFP-

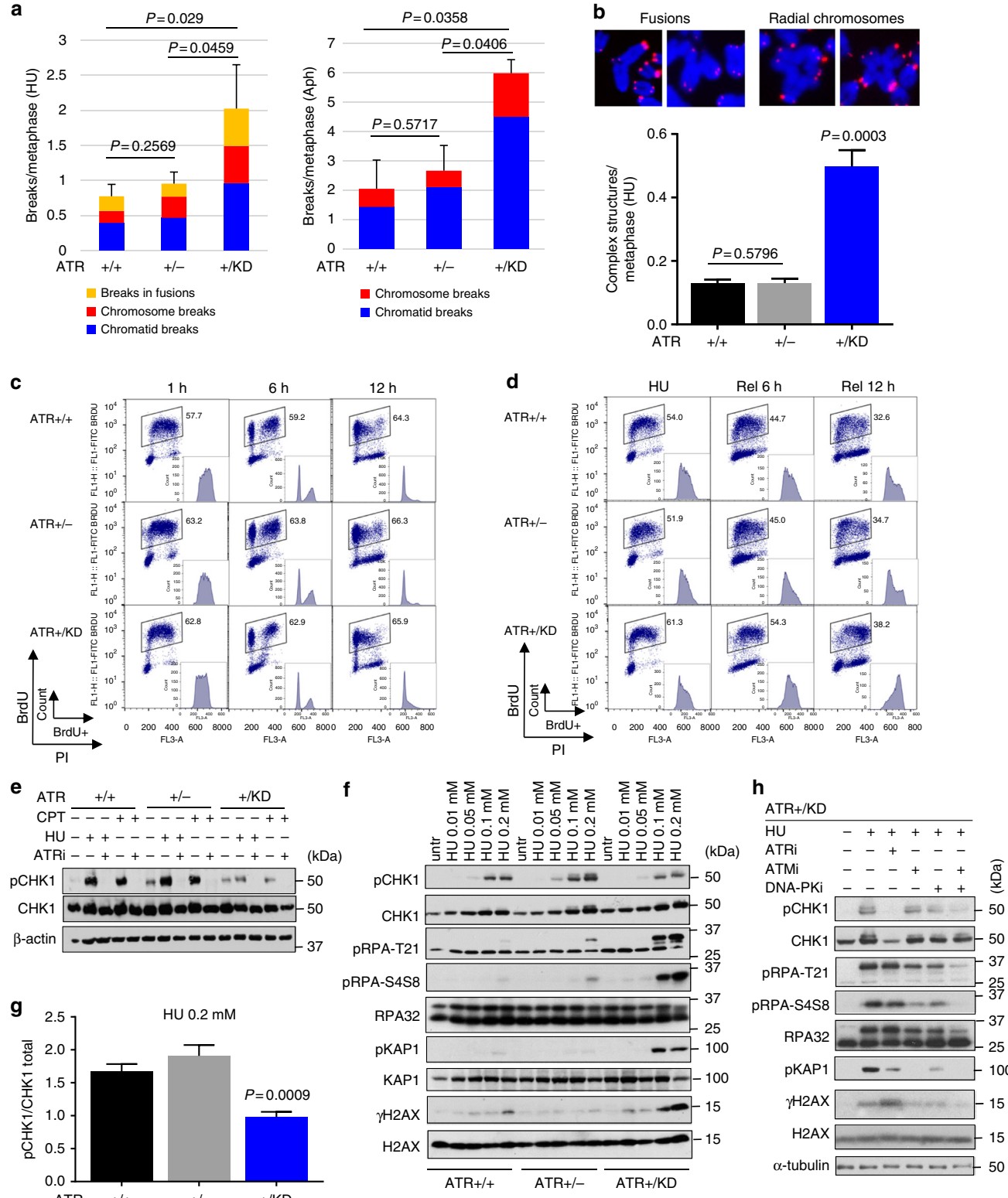

ATR-WT are comparable in the same cell, suggesting that the lack of recovery after photobleaching is likely due to the slow release of the bleached ATR from DNA, not differential recruitment of ATR-KD vs ATR-WT during recovery (Supplementary Figure 4d). Together, these observations demonstrate that ATR exchanges rapidly on RPA-coated ssDNA in a kinase activity-dependent manner.

**$Atr^{+/KD}$ B cells are hypersensitive to replication stresses**. To test whether ATR-KD elicits ssDNA dose-dependent toxicity, we induced ssDNA in replicating cells using HU, which pauses replication forks by depleting nucleotides. HU-induced mitotic catastrophe (defined as >15 cytogenetic abnormalities per metaphase) was observed in ~10% of $Atr^{+/+}$ and $Atr^{+/-}$ B cells, and in more than 20% of $Atr^{+/KD}$ B cells (Supplementary

**Fig. 4** $Atr^{+/KD}$ cells display HU-induced genome instability and checkpoint defects. **a** Metaphases from B cells treated with HU (0.25 mM) or Aphidicolin (0.5 µM) were analyzed for chromatid breaks, chromosome breaks, and breaks in fusions. The bar graphs represent the means ± SD and unpaired two-tailed $t$ test was used. Three (HU) or two (Aphidicolin) independent experiments were performed. A total of 142 ($Atr^{+/+}$), 161 ($Atr^{+/-}$), and 156 ($Atr^{+/KD}$) metaphases in HU, and 83 ($Atr^{+/+}$), 90 ($Atr^{+/-}$), and 91 ($Atr^{+/KD}$) in Aphidicolin were analyzed. **b** The means ± SD of the number of complex rearrangements (fusions, quadri-radials) in HU were plotted from three independent experiments. Unpaired two-tailed $t$ test was used. **c** $Atr^{+/+}$, $Atr^{+/-}$, and $Atr^{+/KD}$ MEFs were pulse-labeled with BrdU for 30 min, washed and released in unchallenged media up to 6 h. Dot plots show the percentage of BrdU-positive cells and histograms show the cell cycle distribution of BrdU-positive cells. **d** $Atr^{+/+}$, $Atr^{+/-}$, and $Atr^{+/KD}$ MEFs were pulse-labeled with BrdU for 30 min, treated with 2 mM HU for 3 h and then released from HU in unchallenged media up to 12 h. Data were plotted as in (**c**). **e** Primary $Atr^{+/+}$, $Atr^{+/-}$, and $Atr^{+/KD}$ MEFs were treated with 2 mM HU for 3 h or 200 nM CPT for 2 h, with or without ATRi (10 µM, VE-821). Whole-cell extracts (WCE) were immunoblotted with the indicated antibodies. **f** $Atr^{+/+}$, $Atr^{+/-}$, and $Atr^{+/KD}$ MEFs were treated with increasing concentrations of HU (0.01, 0.05, 0.1, 0.2 mM) for 1 h, and WCE were immunoblotted with the indicated antibodies. **g** Histogram shows the quantification of pCHK1/CHK1 at 0.2 mM HU derived from three independent experiments. The means ± SD are shown and unpaired two-tailed $t$ test was used. **h** $Atr^{+/KD}$ MEFs were treated with 0.2 mM HU for 1 h with or without pre-treatment for 1 h with the indicated checkpoint inhibitors. ATRi (VE-821) and CHK1i (LY603218) were used at 10 µM concentration, ATMi (KU-55933) at 15 µM, and DNA-PKcsi (NU7441) at 5 µM. WCE were immunoblotted with the indicated antibodies. Source data are provided as a Source Data file

Fig. 5a, b). Overall, >70% of $Atr^{+/KD}$ B cells displayed cytogenetic abnormalities vs 40% and 45% of $Atr^{+/+}$ and $Atr^{+/-}$ cells, respectively (Supplementary Fig. 5a), consistent with hypersensitivity to HU documented in human cells with ectopic expression of ATR-KD[28]. Aphidicolin, which generates ssDNA by stalling replicative DNA polymerases, also induced higher levels of genomic instability in $Atr^{+/KD}$ cells (Supplementary Fig. 5a). Specifically, the frequency of breaks and complex rearrangements (e.g., fusions and quadri-radials) was significantly increased in $Atr^{+/KD}$ cells treated with HU (Fig. 4a, b) or Aphidicolin (Fig. 4a). Most significant increased chromosomal abnormalities in HU or Aphidicolin-treated $Atr^{+/KD}$ cells were chromatid breaks, suggesting a replication origin (Fig. 4a, b). Together with the normal proliferation and lack of spontaneous chromosomal breaks in $Atr^{+/KD}$ cells (Supplementary Figure 5a), these findings support a ssDNA selective toxicity by ATR-KD.

**Checkpoint alterations in $Atr^{+/KD}$ MEFs.** We next examined the impact of ATR-KD expression on the cellular response to replication stress in murine embryonic fibroblasts (MEFs). As in B cells, both primary and immortalized $Atr^{+/KD}$ MEFs proliferated well and displayed normal S phase progression as measured by BrdU pulse-chase (Fig. 4c, Supplementary Fig. 5c and 5d). However, when challenged with HU, $Atr^{+/KD}$ MEFs, unlike $Atr^{+/-}$ and $Atr^{+/+}$ cells, failed to resume cell cycle progression and displayed an extended G2/M arrest lasting at least 12 h, likely caused by the hyperactivation of ATM and DNA-PKcs (see below, Fig. 4d). Despite the presence of WT-ATR, CHK1 phosphorylation induced by 2 mM HU or 200 nM camptothecin (CPT), was markedly reduced in $Atr^{+/KD}$, but not $Atr^{+/-}$, cells, suggesting that ATR-KD impairs checkpoint activation in a dominant-negative manner (Fig. 4e). At low HU doses (0.1–0.2 mM), while CHK1 phosphorylation was still significantly defective in $Atr^{+/KD}$ cells (Fig. 4f, g), RPA was markedly hyper-phosphorylated at T21 and S4/S8 residues in $Atr^{+/KD}$ MEFs and not in $Atr^{+/+}$ or $Atr^{+/-}$ MEFs (Fig. 4f and Supplementary Fig. 5e). CPT also induced hyper-phosphorylation of RPA in $Atr^{+/KD}$ cells (Supplementary Fig. 5f) and the hyper-phosphorylation was not affected by p53 status (Supplementary Fig. 5g). Under the same HU-treated conditions, an ATR-specific inhibitor also abolished CHK1 phosphorylation and induced RPA hyper-phosphorylation in both $Atr^{+/+}$ and $Atr^{+/-}$ cells, suggesting this phenotype is caused by the presence of kinase-dead ATR (Supplementary Fig. 5g). Concomitant treatment with ATM (KU-55933) and DNA-PKcs (NU7441) inhibitors completely abolished the T21 and S4/S8 RPA phosphorylation in HU-treated $Atr^{+/KD}$ cells (Fig. 4h), indicating that ATM and DNA-PKcs are

activated and phosphorylate RPA. Indeed, phosphorylation of KAP1, a specific substrate of ATM, and histone H2AX, a shared substrate of ATM and DNA-PKcs, were also increased in HU-treated $Atr^{+/KD}$ cells and abrogated by ATM and DNA-PKcs inhibition (Fig. 4f, h). In this context, both alkaline and neutral comet tail moments were significantly higher in $Atr^{+/KD}$ cells and remained high after HU release (Fig. 5a and Supplementary Fig. 6a), suggesting frequent and persistent DNA breaks activate ATM and DNA-PKcs in HU-treated $Atr^{+/KD}$ cells. Phosphorylation of CHK1, already compromised in $Atr^{+/KD}$ cells, was also further reduced by both ATM and DNA-PKcs inhibition (Fig. 4h), consistent with the compensatory phosphorylation of CHK1 by DNA-PKcs observed in ATR inhibitor-treated human cells[46]. The relatively low abundance of DNA-PKcs in mouse cells compared to human cells likely explains the evident contribution of ATM observed in murine cells here (Fig. 4h).

**Decreased RPA mobility in ATR kinase-defective cells.** ATM/DNA-PKcs activation is necessary, but not sufficient for RPA phosphorylation. Only chromatin (DNA) bound RPA is phosphorylated in vivo. Using a recently developed flow cytometry-based method[47], we found that low-dose HU induced much higher levels of chromatin-bound RPA in $Atr^{+/KD}$ cells than in $Atr^{+/-}$ or $Atr^{+/+}$ controls (Fig. 5b, c). ATM and/or DNA-PKcs kinase inhibitors abolished HU-induced RPA hyper-phosphorylation in $Atr^{+/KD}$ cells, while did not affect the levels of chromatin-bound RPA (Supplementary Fig. 6b), suggesting the increased chromatinization of RPA is not a consequence of ATM and DNA-PKcs activation. Moreover, chromatin fractionation also showed a greater increase of RPA and phosphorylated RPA (T21 and S4/S8) in the chromatin fraction of HU-treated $Atr^{+/KD}$ cells specifically (Fig. 5d), suggesting that limited kinase activity and exchange of ATR affect the chromatin retention of RPA. To determine whether catalytically-inactive ATR can directly affect RPA exchange at the same DNA lesion, we measured FRAP of GFP-RPA, at 10 min after micro-irradiation. Damage-induced GFP-RPA foci recovered rapidly, even faster than ATR ($t_{1/2} = 5.71 \pm 0.29$ s for RPA vs $13.36 \pm 0.98$ s for ATR) and completely ($103 \pm 3.9\%$) after focal photo-bleaching at the damaged area (Fig. 5e, f). ATR inhibition significantly reduced the maximal recovery of GFP-RPA at the damage sites from $103 \pm 3.9\%$ to $89 \pm 3.9\%$, while CHK1 inhibition did not measurably alter the recovery of RPA (Fig. 5e–g), suggesting ATR can directly modulate RPA exchange independent of CHK1 mediated replication stress responses. The $t_{1/2}$ was not affected by either ATR or CHK1 inhibition ($6.04 \pm 0.89$ and $4.94 \pm 0.39$ s, respectively). ATR inhibition also did not affect the exchange of free RPA not at a DNA

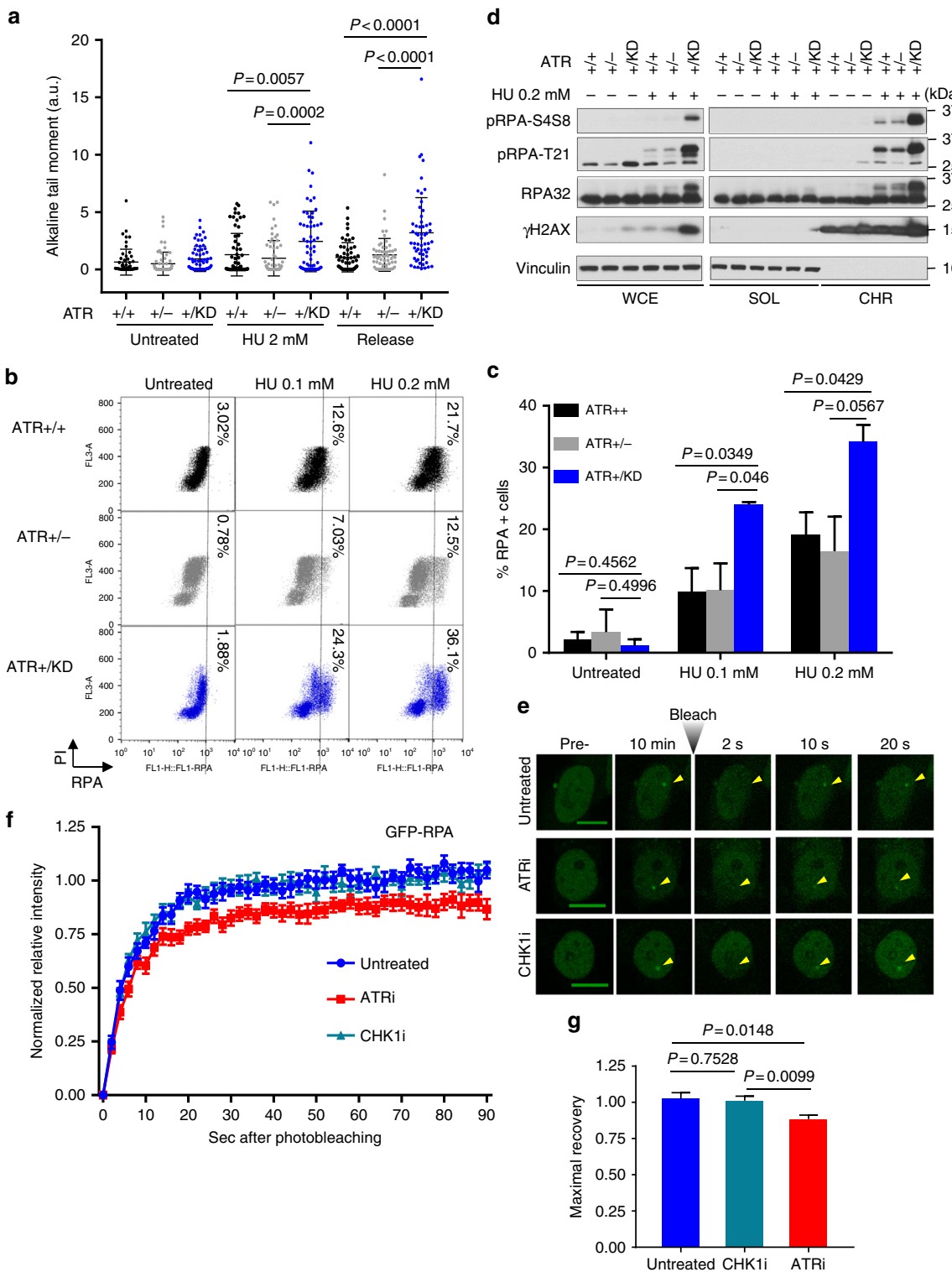

damage site (Supplementary Fig. 6c), suggesting that this effect of ATR-KD on RPA exchange is ssDNA-dependent and direct (not via CHK1). Notably, the exchange of RPA at the damage foci is much faster than that of ATR (with or without ATR inhibition), suggesting the majority of RPA can be exchanged independent of ATR mobility. Consistent with this notion, ATR inhibition reduced the exchange of RPA by ~10%, while reducing the exchange of ATR by 60% (Fig. 3b). One possible explanation to this observation is that only a small subset of RPA-coated ssDNA is occupied by ATR-ATRIP at any given time, since RPA-coated

ssDNA also serves as a platform for the recruitment of other repair proteins (e.g., RAD52, RAD51, SMARCAL1)[48–50].

**ATR-KD exacerbates replication stress-induced fork stalling.** We then tested whether the decreased mobility of both ATR and a subset of RPA on ssDNA prevents repair and recovery after transient replication stress in $Atr^{+/KD}$ MEFs. Consistent with the higher level of DNA breaks observed in $Atr^{+/KD}$ cells after release from HU exposure (Fig. 5a and Supplementary Fig. 6a),

**Fig. 5** Hyperactivation of ATM and DNA-PKcs, increased breaks and chromatin-bound RPA in HU-treated $Atr^{+/KD}$ MEFs. **a** Dot plot of alkaline comet tail moments of $Atr^{+/+}$, $Atr^{+/-}$, and $Atr^{+/KD}$ cells left untreated, treated with 2 mM HU for 3 h or released from HU in unchallenged medium for an additional 3 h. The means ± SD are shown in the dot plot of one representative experiment and unpaired two-tailed $t$ test was used for the statistical analysis. **b** The chromatin-bound RPA fractions are shown as dot plots. The dotted lines were arbitrarily set at $10^3$ and mark the separation between the RPA chromatin-bound positive (>$10^3$) and negative (<$10^3$) cells. The percentages of RPA positive cells from a representative experiment are shown. **c** The mean values ± SD of chromatin-bound RPA in untreated and HU (0.1 mM or 0.2 mM for 1 h) treated cells is reported from two independent experiments. Unpaired two-tailed $t$ test was used. **d** Chromatin fractionation experiment was performed on $Atr^{+/+}$, $Atr^{+/-}$, and $Atr^{+/KD}$ MEFs untreated or treated with 0.2 mM HU. The whole cell extracts (WCE), the soluble (SOL), and the chromatin (CHR) fractions were analyzed by SDS-PAGE and Western blot with the indicated antibodies. **e, f** FRAP analysis of GFP-RPA transfected U2OS cells after laser-induced foci formation, untreated or following treatment with 10 μM ATRi (VE-821) or Chk1i (LY2603618). Representative cells are shown up to 20 s after photobleaching. Scale bar is 10 μm. Recovery of the focus signal was monitored up to 90 s after photobleaching. **g** The means ± SEM of the maximal recovery of GFP-RPA are reported for every condition analyzed and unpaired two-tailed $t$ test was used for the statistical analysis. Source data are provided as a Source Data file

H2AX and RPA phosphorylation levels were also much higher in $Atr^{+/KD}$ cells relative to $Atr^{+/+}$ and $Atr^{+/-}$ cells after HU release (Fig. 6a). However, despite these high levels of H2AX and RPA phosphorylation in $Atr^{+/KD}$ cells, there were significantly fewer RAD51 foci in HU-treated $Atr^{+/KD}$ MEFs than in $Atr^{+/-}$ or $Atr^{+/+}$ controls, suggesting the stalled ATR-KD and increased chromatin retention of ATR-RPA after HU treatment might physically prevent RAD51 from loading onto the RPA/ssDNA (Fig. 6b). Consistent with these data, replication fork recovery, as measured by single molecule DNA fiber analysis, was also significantly lower in HU-treated $Atr^{+/KD}$ cells than $Atr^{+/+}$ and $Atr^{+/-}$ controls (Fig. 6c). Notably, the frequency of RAD51 foci induced by IR, which primarily generates DSBs and activates ATM and DNA-PK, is not significantly affected in $Atr^{+/KD}$ cells (Supplementary Fig. 6d), highlighting the importance of ATR exchange in response to HU-induced replication stress. To understand whether ATR-KD also blocks normal replication under the undisturbed growth conditions, we measured inter-origin distance using molecular combing. Interestingly, the inter-origin distances in $Atr^{+/KD}$ MEFs were much shorter than those from $Atr^{+/+}$ and $Atr^{+/-}$ (Fig. 6d), suggesting increased origin firing. Together, these data indicate that the reduced exchange of ATR-KD protein and a subset of RPA during replication stress response impairs checkpoint activation, limits RAD51 loading, accelerates fork collapses, and slows replication recovery (Fig. 6e). Under basal conditions, frequent firing of dormant replication origins might compensate for these defects and support largely normal proliferation of $Atr^{+/KD}$ cells while creating potential vulnerability to additional DNA damage. Taken together, our data support a model in which kinase-dependent exchange of ATR is limited by ATR inhibition and has much more severe consequences than the loss of ATR.

## Discussion

Here, we provide compelling evidence that ATR inhibition differs from ATR loss in vivo with impacts under both physiological and stressed conditions. Specifically, our data identified ssDNA, but not DSBs, as a platform to reveal the selective toxicity of the ATR-KD protein and further argue that the duration or level of given ssDNA segment matter. Hematopoiesis and lymphocyte development, which involve programmed DSB formation, are largely normal in $Atr^{+/KD}$ mice since joining of such breaks by classical non-homologous end-joining generates none or limited ssDNA. Meiotic recombination at autosomes involving transient ssDNA formation that is quickly replaced by RAD51/DMC1, is largely normal in $Atr^{+/KD}$ mice, but the resolution of X–Y bodies where persistent ssDNA accumulates at the non-homologous regions is abrogated[33]. Similarly, proliferation and S phase progression involving transient ssDNA at the lagging strands that are promptly replaced by ongoing lagging strand synthesis, are well

tolerated in $Atr^{+/KD}$ cells, but genomic instabilities are observed at telomeres, rDNA repeats, and fragile sites, where ssDNA intermediates might accumulate during DNA replication and repair. Notably, we use the word transient here to describe the short-lived nature of specific ssDNA segments, not the time frame during which the cell as a whole has ssDNA at any genomic locations. Meiotic cells or replicating cells can often have ssDNA and RPA-coated ssDNA through meiosis or during the entire S phase, albeit at different locations at different time. Using FRAP, our findings suggest that this ssDNA-dependent toxicity is in part caused by the lack of kinase-dependent rapid exchange of ATR in $Atr^{+/KD}$ mice. As such, the presence of ATR-KD, but not the partial loss of ATR, directly blocks downstream repair, including the exchange of some RPA on chromatin, as well as indirectly precludes proper activation of WT-ATR and blocks DNA repair and checkpoint activation, leading to ssDNA-dependent toxicity in $Atr^{+/KD}$ mice.

HU or Aphidicolin induce extended RPA-coated ssDNA and activate ATR dependent checkpoint. HU-induced Chk1 phosphorylation is compromised in $Atr^{+/KD}$ cells, suggesting the rapid exchange of ATR is necessary for robust CHK1 phosphorylation and DNA damage response, potentially by allowing multiple rounds of ATR activation on limited RPA-coated ssDNA. Moreover, ATR mobility on DNA is also necessary for the exchange of a small yet consistent subset of RPA molecules on chromatin and subsequent repair (e.g., RAD51 foci formation). The hyper-phosphorylation of RPA and hypo-phosphorylation of CHK1 in $Atr^{+/KD}$ cells highlight different requirement for RPA and CHK1 phosphorylation. While CHK1 phosphorylation is mediated predominantly by ATR, RPA phosphorylation requires its chromatin binding and is mostly mediated by ATM and DNA-PKcs. While reduced phosphorylation of CHK1 and other substrates contributes to the overall phenotype of the $Atr^{+/KD}$ mice, neither $Atr^{+/-}$ nor $Chk1^{+/-}$ mice display the ssDNA selective spermatogenesis defects and the rDNA/telomere instability, suggesting the presence of ATR-KD structurally impairs genomic stability beyond simple loss of ATR kinase. The kinase activation dependent exchange of ATR provides a mechanism for the dominant-negative structural function of ATR-KD.

How does ATR kinase activity affect the dynamic exchange of ATR? Structural studies of Mec1-Ddc2 (the yeast homolog of ATR-ATRIP) suggest that the kinase domain of ATR adopts an intrinsically active configuration in which the catalytic loop is enclosed by its neighboring PRD and FAT domains in a manner that occludes substrate access[29]. Enzymatic activation is achieved by the interaction of the PRD domain with ATR activator proteins (e.g., TopBP1 or ETAA1), which elicit significant conformational changes that expose the catalytic loop. A similar allosteric model of PRD-mediated enzymatic activation has been proposed for other PI3KKs, such as ATM and mTOR[29,51,52]. Given RPA binding might allosterically activate ATR, it is also

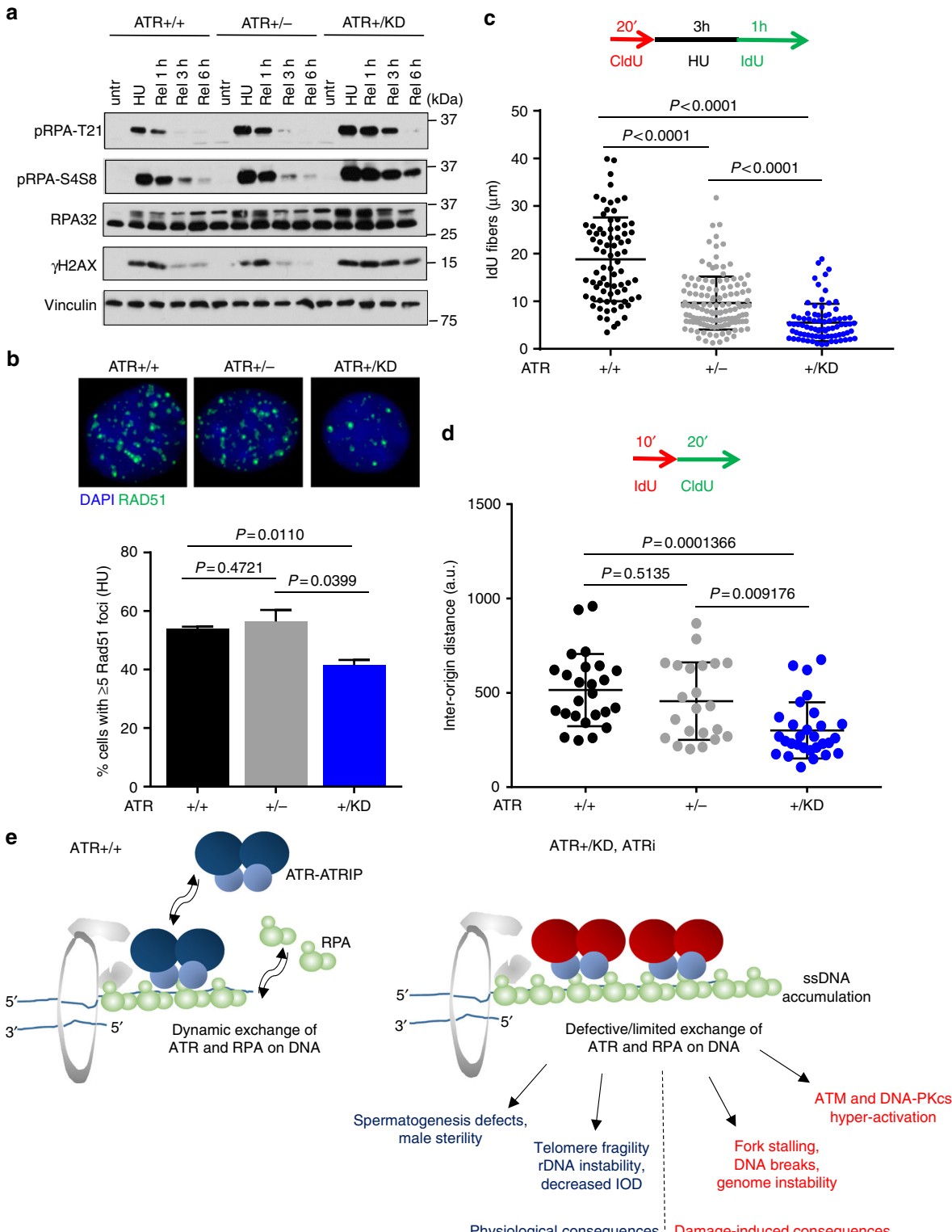

possible that the activated ATR adopts an open and high-affinity RPA-ssDNA binding conformation. As such the catalysis associated conformational changes might regulate the release of ATR from ssDNA/RPA allosterically by reversing the activation process. Although the impaired Chk1 phosphorylation and persistent RPA phosphorylation seen in $Atr^{+/KD}$ cells share similarity with the HU-treated human cells expressing the ATR-T1989A mutant lacking the auto-phosphorylation site[53], T1989 is not strictly conserved in murine ATR and auto-phosphorylation has not

been detected in murine cells yet. Moreover, a structural study on the yeast ortholog of ATR- Mec1, suggests T1989 auto-phosphorylation site is not readily accessible by the catalytic center of ATR or its dimeric partner[29]. In this context, for the related ATM kinase and DNA-PKcs, the expression of kinase-dead protein has much more severe consequences than the loss of clusters of auto- and trans-phosphorylation sites[24–26,54–56], suggesting activation associated conformation changes might occur independent of auto- or trans-phosphorylation.

**Fig. 6** ATR-KD protein exacerbates replication stress-induced replication fork stalling and collapse. **a** $Atr^{+/+}$, $Atr^{+/-}$, and $Atr^{+/KD}$ MEFs were treated with 2 mM HU for 3 h and then released in unchallenged media up to 6 h and the indicated time points (1, 3, and 6 h after HU release) were collected for whole-cell extracts (WCE), that were immunoblotted with the indicated antibodies. **b** $Atr^{+/+}$, $Atr^{+/-}$, and $Atr^{+/KD}$ MEFs were treated with 2 mM HU for 6 h and then released overnight in unchallenged media. Nuclei were stained for RAD51 and DAPI. Data are representative of two independent experiments and a total of about 1000 cells were analyzed for every genotype. The mean values ± SD are shown and unpaired two-tailed $t$ test was used. Representative immunofluorescence images are shown for $Atr^{+/+}$, $Atr^{+/-}$, and $Atr^{+/KD}$ MEFs. **c** $Atr^{+/+}$, $Atr^{+/-}$, and $Atr^{+/KD}$ MEFs were pulse-labeled with 25 μM CldU for 20 min, treated with 2 mM HU for 3 h and then released in media containing 125 μM IdU for 1 h. IdU track lengths were measured using ImageJ software. Means ± SD are shown in the dot plot of one representative experiment and Mann–Whitney test was used for the statistical analysis. **d** Molecular combing experiment in $Atr^{+/+}$, $Atr^{+/-}$, and $Atr^{+/KD}$ MEFs in unchallenged conditions. Cells were labeled with IdU for 10 min and then with CldU for 20 min. Inter-origin distances were measured and statistical analysis was done with ANOVA test. Means ± SD are shown in the dot plot. **e** Model of the function of ATR kinase activity on ATR and RPA exchange describing the dominant-negative effect of ATR-KD protein in physiological conditions (meiosis, telomere, and rDNA replication) and upon replication stress challenge. Source data are provided as a Source Data file

Given the strong dominant-negative effect of ATR-KD documented in previous overexpression studies[31], it is surprising that $Atr^{+/KD}$ mice are born of normal size and fertile (female) and $Atr^{+/KD}$ cells proliferate efficiently in culture. While it is possible that the rapid replacement of RPA on ssDNA, such as during ongoing lagging strand synthesis, might un-root ATR (and ATR-KD) and explain why ATR-KD does not severely impair cell proliferation, we also found that $Atr^{+/KD}$ cells have shorter inter-origin distance, which might compensate for the frequent stalling and reflect a compromised basal ATR activity in $Atr^{+/KD}$ cells. In this case, it is tempting to speculate that the frequent firing of dormant origins might also contribute to the genomic instability in $Atr^{+/KD}$ cells at the telomere and rDNA loci, and the vulnerability of $Atr^{+/KD}$ cells to exogenous genotoxic agents. The accumulation of these mild, yet consistent genomic instabilities throughout the numerous rounds of proliferation required for lymphocyte development and maturation, might underlie the lymphocytopenia and B cell maturation defects in $Atr^{+/KD}$ mice.

Taken together, our data show that kinase-dependent exchange of active ATR at ssDNA/RPA filaments, while dispensable for normal development and general mitosis, is necessary for robust DNA damage response, male-specific meiosis and genomic stability at telomere, rDNA and fragile sites. Furthermore, the FRAP data show a direct effect of catalytic inactive ATR on the exchange of ATR itself and a subset of RPA. Our finding supports a model in which ATR kinase activity regulates the exchange of ATR itself both directly (through trapping) and indirectly (through reduced ATR activity) to increase chromatin retention of RPA and affect the downstream repair. The selective and ssDNA-dependent toxicity of ATR-KD predicted by this model provides an uniform explanation for the low toxicity of ATR inhibition in normal cells, where the limited ssDNA/RPA tracts generated during DNA replication would be insufficient to impair viability, and the synergistic effects elicited by accumulating R-loops[57,58], and chemotherapeutic agents that generate extensive ssDNA, such as cisplatin, but not radiation, which primarily produce DSBs. Therefore, our findings suggest extended ssDNA as a quantitative biomarker for therapy involving ATR inhibition.

## Methods

**Generation of ATR kinase-dead mouse model.** To generate the ATR KD (D2466A) allele, a 4.9 kb 5′ arm and 3.5 kb 3′ arm from the ATR locus were PCR amplified from CLS3 ES cell DNA (129 strain), cloned into pBluescript (pBK) shuttle vector and sequenced. The mutation was introduced into the 5′ arm using site-directed mutagenesis and confirmed by sequencing. The mutated 5′ arm (indicated in Supplementary Fig. 1a) was then sub-cloned into pEMC vector with NeoR gene cassette flanked by two FRT sites. The targeting construct was then electroporated into CSL3 ES cells (129 strain) and successful targeting was determined via Southern blot analyses using StuI digested genomic DNA and a 3′ genomic probe as outlined in Supplementary Fig. 1a (Germline = 6.9 kb and targeted = 4.8 kb) with 3′ probe. The correct clones were confirmed with HindIII digestion (Germline = 13.4 kb and targeted = 11.2 kb) with 5′ probe and the mutation was verified by Sanger sequencing. The correct targeting clones contain

the NeoR in the opposite transcriptional orientation from the endogenous $Atr$ promoter in intron 44 next to the D2466A mutation in exon 44. A total of 6 correctly targeted clones were identified and two were injected for germline transmission. The resulted chimers were crossed with germline ROSA[FLIP/FLIP] mice (Jax Laboratory, Cat. 003946) to remove the FRT-NeoR-FRT cassette and obtain the $AtrKD$ (kinase-dead) allele. Southern blotting was performed following standard protocol.

All the mice used in the study are of 129/Sv background. All animal experiments were conducted in a pathogen-free facility at the Institute of Cancer Genetics (Columbia University Medical Center, CUMC) in accordance to all relevant ethical regulations and approved by the Institutional Animal Care and Use Committee (IACUC) of Columbia University.

**ATR alleles used in the study.** The $Atr^C$ and $Atr^-$ alleles used to derive the MEFs were generously provided by Dr. Eric Brown[32,59]. Briefly, $Atr^-$ allele was generated by deleting the first three exons that encode the first methionine and 90 amino acids of murine ATR. $Atr^C$ allele includes a pair of LoxP sites flanking the two exons encoding the core kinase domain of murine ATR.

**Lymphocyte development and class switch recombination.** Single-cell suspensions were prepared from lymphocytes of thymus (T), bone marrow (BM), spleen (Spl), and lymph node (LN) from mice of the described genotypes and $\sim 1 \times 10^5$ cells in 1× PBS were stained for 15 min at RT using fluorescence-conjugated antibodies and analyzed by flow cytometry[60]. The following antibodies were used: FITC-Cd11b (1:200, M1/70, BD Pharmingen, 553310), APC-Ly-6G/Ly-6C (Gr-1) (1:200, RB6-8C5, Biolegend), PE-IgM (1:400, SouthernBiotech, 1020-09), PE-CD4 (1:200, GK1.5, BD Pharmingen, 553730), FITC-CD8a (1:200, 53-6.7, Biolegend), PE-Cyanine5-CD3e (1:200, 145-2C11, Invitrogen, 15-0031-63).

For CSR assay, single-cell suspensions of spleen cells were sorted with CD43 magnetic beads (MACS, Miltenyi), and CD43- B cells were cultured at a density of $5 \times 10^5$ cells per ml in RPMI medium supplemented with 15% FBS and 25 ng ml$^{-1}$ of IL-4 (R&D) and anti-CD40 (BD Biosciences). Cultured cells were maintained daily at a density of $1 \times 10^6$ cells per ml. Cells were collected every day up to day 4.5 and were stained with FITC-conjugated IgG1 (1:200, A85-1, BD Pharmingen, 553443) and PE-Cyanine5-conjugated B220 (1:200, RA3-6B2, BD Pharmingen, 553091). Flow cytometry was performed on a FACSCalibur flow cytometer (BD Biosciences) and data were processed using FlowJo software package.

**TUNEL assay.** TUNEL assay was performed on paraffin-embedded testis sections using the In Situ Cell Death Detection Kit (Roche, 11684817910) and following manufacturer instructions.

**Drugs and inhibitors.** The following drugs and checkpoint inhibitors were used: Hydroxyurea (Calbiochem), Aphidicolin (Acros Organics), Camptothecin (Calbiochem), ATRi VE-821 (Selleckchem), CHK1i LY2603618 (Selleckchem), ATMi KU-55933 (Selleckchem), DNA-PKcsi NU7441 (Selleckchem).

**MEFs cultures, cell growth assay, and cell cycle analysis.** Mouse embryonic fibroblasts (MEFs) were harvested at day 14.5 (E14.5) based on timed breedings. Primary MEFs were used at early passages (P1–P3). Immortalized MEFs were obtained through retrovirus expression of the large and small SV40 T antigen. Immortalized cultures were established after a few passages from the primary infection. MEFs were cultured in DMEM supplemented with 15% FBS, glutamine, sodium pyruvate, and penicillin/streptomycin. For cell growth assay, primary MEFs were seeded in gelatinized 96-well plates ($10^3$ cells/well). The cell number was quantified at the indicated days using the CyQuant DNA stain (Invitrogen) following the manufacturer's instructions using a fluorescence microplate reader (GloMax, Promega). The relative survival was normalized to the cell number at day 1 (24 h after plating, which is indicated as day 0 in the graph in Supplementary Figure 5c).

For cell cycle analyses, proliferating primary or immortalized MEFs were incubated with 10 μM of 5-bromo-2′-deoxyuridine (BrdU, Sigma) for 30 min and treated as indicated. Cells were fixed in 70% ethanol overnight, permeabilized, and DNA denatured with acid solution (2 N HCl, 0.5% Triton X-100) for 30 min at RT and washed with sodium phosphate citrate buffer pH 7.4. Cells were then stained with FITC-conjugated mouse anti-BrdU antibody (1:10, BD Pharmingen, 556028) for 30 min. Finally, cells were incubated with propidium iodide (Sigma) and RNase A (Sigma) for additional 30 min in the dark. More than 25,000 cells were acquired on a FACSCalibur flow cytometer (BD Biosciences) and analyzed by FlowJo software package.

**Protein extraction, antibodies used, and western blot.** Whole cell extracts (WCE) were prepared using RIPA buffer (150 mM NaCl, 10 mM Tris–HCl pH 7.4, 0.1% SDS, 0.1% Triton X-100, 1% sodium deoxycholate, 5 mM EDTA) supplemented with 2 mM PMSF, 10 mM NaF, 10 mM β-glycerophosphate, and protease inhibitor cocktail (Roche). SDS-PAGE and western blots were performed following standard protocols. Primary antibodies used in the study are: pKAP1 S824 (1:1000, A300-767A, Bethyl Laboratories), KAP1 (1:1000, TIF1β, C42G12 Cell Signaling, #4124), pCHK1 Ser245 (1:1000, 133D3 Cell Signaling, #2348), CHK1 (1:1000, 2G1D5 Cell Signaling, #2360), pRPA32 T21 (1:10000, Abcam, ab109394), pRPA S4/S8 (1:3000, A300-245A, Bethyl Laboratories), RPA32 (1:10000, A300-244A, Bethyl Laboratories), pH2AX Ser139 (1:1000, 07-164, Millipore), H2AX (1:1000, 07-627, Millipore), ATR (1:1000, Cell Signaling, #2790), vinculin (1:1000, V284, 05-386 Millipore), β-actin (1:1000, AC-15, A1978 Sigma), α-tubulin (1:1000, CP06, Calbiochem). Western blots from at least three independent experiments were quantified using ImageJ. All uncropped western blots from the main figures are shown in Supplementary Figures 7.

**Chromatin fractionation.** Cells in the mid-exponential phase of growth were washed once with 1× phosphate-buffered saline (PBS) and lysed in ice-cold lysis buffer A (20 mM Tris–HCl pH 7.5, 10 mM KCl, 1.5 mM MgCl₂, 0.1% Triton X-100) containing protease inhibitor cocktail (Roche) for 10 min on ice. Nuclei were collected by low-speed centrifugation (1300×g, 5 min at 4 °C). The supernatant was clarified by high-speed centrifugation (15,000×g, 5 min at 4 °C) and was collected as cytosoluble fraction. The nuclei were washed once with lysis buffer A and lysed with buffer B (300 mM NaCl, 10% glycerol, 20 mM Tris–HCl pH 7.9, 1 mM β-mercaptoethanol, 0.1% Triton-X-100) supplemented with protease inhibitor cocktail (Roche) and 1 mM PMSF for 30 min on ice. Low-speed centrifugation (1300×g, 5 min at 4 °C) was performed to separate the nucleosoluble fraction and the chromatin-bound proteins (pellet). The pellet was washed once in buffer B, resuspended in SDS-Laemmli buffer, and boiled for 10 min before SDS-PAGE. For each fraction obtained, protein amounts from a comparable number of cells were then analyzed by SDS-PAGE and western blotting.

**Quantification of chromatin-bound RPA by flow cytometry.** The recruitment of RPA to chromatin was performed essentially as described[47] with minor modifications. This protocol allows to quantify only the fraction of RPA that is associated to chromatin and to measure in parallel the cell cycle distribution using propidium iodide (PI) staining. Briefly, untreated or HU-treated cells were permeabilized with PBS/Triton-X 0.2% for 5 min on ice (non-chromatin-bound protein extraction step), washed with BSA 5%/PBS and then fixed with paraformaldehyde 4% for 10 min at room temperature. Cells were then resuspended in BSA 5%/PBS for 1 h, incubated with primary RPA32 antibody (1:100, NA19L, Millipore) for 2 h at room temperature and then with Alexa Fluor 488-conjugated secondary goat anti-mouse antibody (Thermo Fisher Scientific) for 1 h in the dark. Finally, DNA was stained with PI (Sigma) for 20 min at room temperature in the dark. More than 25,000 cells for every sample were acquired using CellQuest software on FACSCalibur instrument and analyzed using FlowJo.

**Metaphase spreads and telomere-FISH (T-FISH).** Metaphase spreads were collected from in vitro activated B cells at day 4.5 after stimulation. B cells were left untreated or treated at day 3.5 with HU (250 μM) or Aphidicolin (0.5 μM) overnight (18–20 h). At the day of collection, cells were treated with Colcemid (KaryoMax, Gibco) at 100 ng/ml final concentration for 2 h, harvested with hypotonic KCl solution and fixed with cold methanol/acetic acid (3:1) solution. The fixed suspension was dropped onto precleaned superfrost microscope slides (Fisher Scientific) to obtain chromosome spread and left to dry overnight. Slides were fixed in 4% (w/v) formaldehyde/PBS for 2 min, followed by three washes with PBS and digestion with pepsin/PBS (0.1%; Sigma) for 10 min at 37 °C. The slides were then washed in PBS twice, fixed in 4% (w/v) formaldehyde/PBS for 2 min, washed again in PBS three times, and dehydrated through ethanol 70%, 90%, and 100%. A Cy3-labeled (CCCTAA)3 peptide nucleic acid (PNA) probe (customer synthesized, Biosynthesis Inc.) was used to hybridize the metaphases under denaturing conditions (heating for 3 min at 80 °C) and incubated in a dark humidity chamber for at least 2 h at 37 °C. The slides were then washed with 70% high-purity formamide, 10 mM Tris–HCl pH 7.2, 0.1% BSA, twice for 15 min each. Finally, slides were washed three times with PBS Tween 0.08%, dehydrated in ethanol 70%, 90%, and 100% and air dried. DNA was counterstained with Vectashield containing DAPI (Vector Laboratories). Metaphases were scanned using Carl Zeiss AxioImager Z2

equipped with a CoolCube 1 camera and a 63×/1.30 oil objective. Metafer4 software (MetaSystems) was used and images were analyzed and exported using the ISIS fluorescence imaging platform (MetaSystems). Data are representative of two or three independent experiments.

**FISH at GIMAP fragile site.** Two BACs, RP24-371A23 and RP24-387K23, containing the GIMAP locus on chromosome 6, were purchased from BACPAC Resources Center (BPRC). After purification, BACs were labeled by nick translation using DIG-11-dUTP, DNA Polymerase I and DNase I at 16 °C overnight[61]. DNA length was verified to be around 500–900 bp after nick translation and the reaction was stopped using 0.5 M EDTA at 65 °C for 10 min. After DNA precipitation, 100 ng of labeled probe was used for each metaphase spread slide, denatured at 76 °C for 5 min and incubated overnight at 37 °C in a humidified chamber. Slides were washed in pre-warmed (45 °C) 50% formamide/2× SSC for 3 times (5 min each) shaking. Slides were then washed in pre-warmed (45 °C) 0.1× SSC for 3 times (5 min each), dipped in 4× SSC and blocked with Blocking Solution (3% BSA/4× SSC/0.1% Tween 20) at 37 °C for 30 min in a humidified chamber. After quick dipping in 4× SSC, slides were incubated with Alexa 594 conjugated anti-Biotin antibody (Molecular Probe, 1:200 in 1% BSA/4× SSC/0.1% Tween 20) at 37 °C for 45 min in a humidified chamber in the dark. Slides were washed in pre-warmed (45 °C) 4× SSC/0.1% Tween 20 for 3 times (5 min each) shaking, then washed for 5 min in 2× SSC, dipped quickly in dH₂O and finally sealed with Vectashield containing DAPI (Vector Laboratories). Metaphases were scanned using Carl Zeiss AxioImager Z2 equipped with a CoolCube 1 camera and a 63×/1.30 oil objective. Metafer4 software (MetaSystems) was used and images were analyzed and exported using the ISIS fluorescence imaging platform (MetaSystems).

**Murine rDNA FISH.** The single BAC (BK000964) contains one murine rDNA repeat and is approximately 42 kb in length with about 12 kb of which that is transcribed to produce pre-rRNA and 30 kb of intergenic spacer (IGS)[40]. To visualize rDNA by FISH, we used a ~12 kb EcoR I restriction fragment that is positioned in the IGS immediately upstream of the gene promoter, similar to what has been published for human rDNA FISH probe[62]. Since these sequences are mostly non-transcribed, we can be sure that we are visualizing rDNA and not rRNA. This rDNA FISH probe can readily identify individual nucleoli organization regions (NOR) on metaphase spreads and reveal the variation in their rDNA content. The plasmid with the 12 kb RI fragment was generously provided by Dr. Brian McStay at National University of Ireland Galway, Ireland. Standard nick translation labeling was used to label the IGS fragment and FISH was performed as described above using 40 ng of labeled probe for each metaphase spread slide. Metaphases were acquired as described above for telomere-FISH and GIMAP FISH. The number of rDNA clusters was counted for every metaphase analyzed, considering 8 as the physiological number in the 129/Sv mouse strain we used.

**Rad51 immunofluorescence and foci quantification.** Immunofluorescence was performed on cells grown on coverslips, fixed with 4% paraformaldehyde for 10 min and permeabilized with PBS/0.2% Triton-X-100 for 5 min. Cells were then blocked in 5% BSA/PBS for 1 h and incubated with a Rad51 antibody (1:200, PC-130, Calbiochem) for 2 h. After three washes with PBS, cells were incubated with Alexa Fluor 488-conjugated goat anti-rabbit secondary antibody (Thermo Fisher Scientific) for 1 h in the dark. Slides were washed with PBS, dehydrated with ethanol 70%, 90%, and 100% and finally sealed with Vectashield containing DAPI (Vector Laboratories). Slides were scanned using the Carl Zeiss Axio Imager Z2 equipped with a CoolCube 1 camera and a 40×/0.75 objective. Metacyte software (MetaSystems) was used for the automated quantification of Rad51 foci. 500 cells were analyzed for every experimental condition. Data are representative of two independent experiments.

**Spermatocyte spreads preparation and immunofluorescence.** Spermatocyte spreads were prepared from mice testes with minor modifications form established protocols[63]. Briefly, tubules were isolated from freshly recovered testes in 1× PBS, incubated with collagenase (1 mg/ml) for 10 min at RT and teared in small pieces in order to release spermatocytes. Recovered spermatocytes were resuspended in hypotonic buffer (30 mM Tris–HCl, 50 mM sucrose, 17 mM trisodium citrate dehydrate) for 30 min at RT, centrifuged and then resuspended in 100 mM sucrose. Cell sucrose suspension was finally dropped on a microscope slide coated with fixative buffer (1% PFA, 0.15% Triton X-100). After spermatocytes were spread and let dry, slides were blocked in 5% BSA/PBS/Tween 0.1% for 1 h and stained with primary antibodies for an additional hour. Primary antibodies used: rabbit SCP3 (1:200, Abcam, ab15093), mouse SCP3 (1:200, Abcam, ab97672), rabbit phospho histone H2AX S139 (1:250, 07-164, Millipore), rabbit RAD51 (1:200, Calbiochem, PC-130), mouse MLH1 (1:200, G168-15, BD Pharmingen, 550838). Secondary antibodies Alexa 488 donkey anti-mouse and Alexa 594 goat anti-rabbit (Thermo Fisher Scientific) were used for 30 min in the dark. Slides were finally sealed with Vectashield containing DAPI (Vector Laboratories). Images were acquired using Carl Zeiss AxioImager Z2 equipped with a CoolCube 1 camera and a 63×/1.30 oil objective. Metafer4 software (MetaSystems) was used and images were analyzed and exported using ISIS for manual analysis.

**DNA fiber**. Exponentially growing MEFs were first incubated with 25 μM IdU (Sigma) for 20 min, washed twice with PBS, treated with 2 mM HU for 3 h, washed again and then incubated with additional 125 μM CldU (Sigma) for 1 h. Cells were then trypsinized and harvested in ice-cold 1× PBS, lysed using spreading buffer (0.5% SDS, 200 mM Tris–HCl pH 7.4, 50 mM EDTA) and stretched along glass coverslips allowing DNA to spread. Slides were air-dried, fixed in methanol and acid acetic (3:1) for 5 min, rehydrated in PBS for 10 min, and denatured with 2.5 M HCl for 30 min at room temperature. Slides were rinsed in PBS and blocked in PBS/0.1%Triton-X-100/5% BSA for 1 h at room temperature. Slides were then stained using primary rat anti-BrdU (1:100, BU1/75 (ICR1), Abcam, ab6326) and mouse anti-BrdU (1:100, B44, Beckton Dickinson, 347580) antibodies for 1 h and then with corresponding secondary goat anti-rat Alexa 594 and anti-mouse Alexa 488 (Thermo Fisher Scientific) antibodies for 30 min. Slides were then mounted in Prolong Gold Antifade reagent (Invitrogen). DNA fibers were analyzed on a Nikon Eclipse 80i microscope equipped with remote focus accessory and a CoolSNAP HQ2 camera unit with a Plan Apo 40×/0.95 objective. All images were processed with NIS-Elements AR software and quantified using ImageJ. Two independent experiments were performed and one representative experiment is plotted.

**DNA combing**. DNA combing was performed using molecular combing system from Genomic Vision, according to the manufacturer's protocol. Cells were sequentially incubated with IdU (10 min) and CldU (20 min) and embedded in 1% LMP agarose plugs. Plugs were digested with Proteinase K for 20 h, melted at 68 °C, and digested with beta-agarase. Silanized coverslips (Genomic Vision) were dipped into the DNA solution and pulled up at 225 μM/s, baked for 2 h at 60 °C, denatured in 1 M NaOH/1.5 M NaCl, washed with PBS, dehydrated with ethanol, and stained with IdU (1:100, B44, Beckton Dickinson, 347580), CldU (1:100, BU1/75 (ICR1), Abcam, ab6326) antibodies and an antibody against single-stranded DNA (1:100, 16-19, MAB3034, Millipore). DNA strands were imaged using a Nikon Ti90 epi-fluorescence microscope (Nikon Inc., Melville, NY, USA).

**Comet assay**. Alkaline and neutral Comet assays were performed following manufacturer instructions (Trevigen). Comets were acquired using a Nikon 80i fluorescence microscope equipped with remote focus accessory and a CoolSNAP HQ2 camera unit with a Plan Apo 20×/0.75 objective. All images were processed with NIS-Elements AR software and tail moments were analyzed using Comet-Score version 1.5. At least two independent experiments were performed for every treatment and one representative experiment is plotted.

**RNA extraction and RT-PCR**. RNA was isolated from B cells using TRIzol (Invitrogen) following manufacturer's protocol. cDNA was generated using SuperScript III First-Strand Synthesis System (Invitrogen) and quantitative PCR was performed using Absolute Blue qPCR SYBER Green ROX Mix (Thermo Fisher Scientific) following the manufacturer's instructions on a 7500 Real-Time PCR System (Applied Biosystem) instrument using the 7500 fast software v2.3. Analysis was based on the comparative $C_t$ method (also known as the $2^{-\Delta\Delta Ct}$)[64].

Oligos used are the following (F 5′-GAGAGGGAAATCGTGCGTGA-3′; R 5′-ACATCTGCTGGAAGGTGG-3′), α-tubulin (F 5′-CCAGATGCCAAGTG ACAAGA-3′; R 5′-GGGTTCCAGGTCTACGAACA-3′); BCL2 (F 5′-GGACTTG AAGTGCCATTGGT-3′; R 5′-AGCCCCTCTGTGACAGCTTA-3′); BACH2 (F 5′-TGAGGTACCCACAGACACCA-3′; R 5′-ACCCTGAGGTACCGTGTGAG-3′); SWAP70 (F 5′-GAGGAGGTTGGCAACAAGAA-3′; R 5′-CCATGCCCTTGCTA AACTGT-3′); GIMAP (F 5′-CTTGGCCAAAGGCATTGTAT-3′; R 5′-GCAGAT GGTGTGACGGTATG-3′).

**RNA-sequencing**. RNA was isolated from testes using TRIzol (Invitrogen) following the manufacturer's protocol. RNA quality was determined using the Bioanalyzer RNA kit (Agilent) and samples with RNA integrity number (RIN) higher than 8 were sent for next-generation sequencing (NGS). TruSeq Stranded mRNA Library Prep Kit from Illumina was used and the library was sequenced with Illumina HiSeq 2000. Analysis was carried out using DESeq and results were reported in FPKM (fragments per kilobase million).

**Fluorescence recovery after photobleaching (FRAP)**. U2OS used were derived from U2OS ATCC HTB-96. U2OS cells were seeded onto glass bottom 35 mm diameter plates at about $10^5$ cells per plate. Transient transfections were performed using Lipofectamine 2000 (Invitrogen) following the manufacturer's instructions and cells were used 48 or 72 h later for experiments. Live cell imaging was carried out using a Nikon Ti Eclipse inverted microscope (Nikon, Inc.) equipped with A1 RMP (Nikon, Inc.) confocal microscope system (Nikon, Inc.) and Lu-N3 Laser Units (Nikon, Inc.). Laser micro-irradiation manipulation and time-lapse imaging were performed with the NIS Element High Content Analysis software (Nikon, Inc.), using a 405 nm laser and energy level that is adequate for ATR and RPA accumulation. A small area of the nucleus was photobleached with a 488 nm laser and fluorescence recovery was captured every 2 s for 90 s. Relative intensity at laser-damaged sites was calculated as the ratio of the mean intensity at each of the micro-irradiation damaged sites to the corresponding mean intensity of the nucleus as background (the entire nucleus was considered). The normalized fluorescence was

then plotted against time after bleach. Fiji software was used for quantifications and a total of at least 10–15 individual cells were analyzed for each data point in at least two independent experiments. The quantification of maximal recovery (or mobile fractions) and $t_{1/2}$ was performed with minor modifications from the original papers as described below[65,66]. The maximal recovery is defined as the difference between the fluorescence intensity in the bleached area before bleaching and the recovered intensity at infinite time (plateau) after bleaching, while $t_{1/2}$ is defined as the time needed for the fluorescence level to reach 50% of the maximum recovered intensity. The statistical analysis of the maximal recovery intensities was performed using unpaired two-tailed $t$ test. In supplementary figure 4d, we fit linear mixed models to evaluate the association between time in seconds and GFP/RFP intensities with random intercepts for cells and bootstrap re-sampling method was used to estimate the 95% confidence interval with 1000-fold replication. GFP-RPA was generously provided by Dr. Cristina Cardoso. RFP-ATR (WT and KD) and GFP-ATRIP were generated from ATR and ATRIP plasmids generously provided by Dr. Lee Zou.

**Statistical analyses**. Statistical analyses were performed using Fisher's exact test, unpaired two-tailed $t$ test, Mann–Whitney test, or ANOVA test as indicated in the figure legends. Microsoft Excel and GraphPad Prism 7 were used for data analyses and plotting.

## Data availability

The authors declare that data supporting the findings of this study are available with the paper and its supplementary information files. The source data underlying Figs. 1b, c, e, f, 2a–e, 3b, c, 4a, b, e–h, 5a, c, d, f, g, 6a–d and Supplementary Figs. 1a, c, d, f, h, 2c, e, f, 3c, d, 4a, c, d, 5a, c, e-g, 6a, c, d are provided as a Source Data file. The RNA-sequencing data are publicly available online under the GEO accession number GSE122139. The original flow cytometry files (FCS) from critical experiments of the main figures have been deposited in Flow Repository and can be retrieved at the following URLs: [http://flowrepository.org/experiments/1919], [http://flowrepository.org/experiments/1920], [http://flowrepository.org/experiments/1921]. All data is available from the authors upon reasonable request.

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

## Acknowledgements

The authors thank Drs. Theresa Swayne and Chyuan-Sheng Lin for technical assistance and advices. The authors thank Dr. Eric Brown for providing mouse model with ATR null and ATR conditional alleles, Dr. Lee Zou and Dr. Alberto Ciccia for providing ATRIP and ATR plasmids, Dr. Cristina Cardoso for sharing the GFP-RPA plasmids, and Dr. Brian McStay for sharing the murine rDNA probe. The authors thank Dr. Elena Raffetti for help with statistical analyses in Supplementary Fig. 4d and Dr. Wei Yang for advising on statistical methods. The authors greatly appreciate the thoughtful revisions by Dr. Richard Baer, and the comments and suggestions by Drs. Lorraine Symington, Jean Gautier, and Rodney Rothstein. The authors apologize to colleagues whose work could not be cited due to space limitations and was covered by reviews instead. This work is in part supported by NIH/NCI 5R01CA158073 and 5R01CA215067 to S.Z. and NIH/NCI 5R01CA204173 to C.J.B. S.Z. is a Leukemia Lymphoma Society Scholar and D.M. is supported by an American–Italian Cancer Foundation Post-Doctoral Research Fellowship.

## Author contributions

D.M., W.J., and S.Z. designed the experiments and interpreted the data. W.J. generated the $Atr^{+/KD}$ mouse model and characterized the spermatogenesis together with D.M. B.J.L. performed the live cell imaging experiments with the help from Z.S. and D.M. D.M. characterized the DNA damage response, DNA replication, and genomic instabilities in the $Atr^{+/KD}$ mice and cells. T.M. and C.J.B. performed and interpreted the inter-origin distance analyses in Fig. 6d. V.E. helped with the maintenance of the $Atr^{+/KD}$ mouse

model colony. M.G.F. collaborated with sensitivity analyses of $Atr^{+/KD}$ cells. D.M. and S.Z. wrote the manuscript.

## Additional information

**Competing interests:** The authors declare no competing interests.

