## [Peer Review File · Nature Communications]

Reviewers' comments:

Reviewer #1 (Remarks to the Author):

The authors of this study generated a mouse model expressing a kinase dead form of ATR, and compare several phenotypes of these ATR+/KD mice to both WT and ATR+/- mice. They show that the ATR+/KD displays male infertility due to defects in spermatogenesis, as well as fragility at telomeres and rDNA loci. The authors link this phenotype to defects in RPA exchange on single stranded DNA, leading to lack of CHK1 phosphorylation. These results further elucidate the effects of ATR inhibition on genome stability, and as such this study should be of interest to the genome instability community. There are however some concerns that need to be addressed before this manuscript is suitable for publication, as outlined below.

-The authors state that Atr+/KD mice display reduced efficiency of class switch recombination (Figure 2C). This is only true compared to WT mice, but not compared to Atr+/- mice. This should be clearly stated in the text.

-In Figure 2D, the authors claim an increased telomere fragility in Atr+/KD cells. Are these differences significant? In either case, (lack of) significance should be clearly indicated in the figure and text.

-The authors show increased fragility at telomeres and rDNA loci in Atr+/KD cells (Figure 2D and 2E), and subsequently claim that Atr+/KD cells show elevated expression of B-cell specific early replicating fragile sites (ERFS). It is unclear why the authors chose to assess expression levels of these genes associated with ERFS (not actual ERFS), instead of DNA breakage at ERFS as shown in the original paper describing ERFS. Furthermore, HU appears to induce a relative 2-2.5x increased expression in Atr+/KD cells for both the ERFS genes and tubulin. Is this effect specific to ERFS?

-Related to the previous point, if the authors can show increased breakage at ERFS, it would be interesting to see if this is also true for common fragile sites, since it is known that ATR regulates CFS breakage.

-They measure rDNA instability by assessing the number of rDNA clusters different from 8. Can they give a reference that states that the physiological number of clusters is 8? Did ATR+/KD lead to greater than or less than 8 cluster or both? What is the interpretation of these findings? Are there breaks associated with rDNA in metaphase spreads?

-They should explain why photobleaching experiments not performed in ATR+/KD cells, but WT cells treated with ATRi.

-Figure 5C, show percentages of RPA+ cells in the left panel, and absolute values instead of fold increases in right panel. Please indicate if differences between genotypes are significant.

Figure 6C, show ratio of IdU/CldU fiber length instead of absolute IdU fiber length.

Figure S5B, indicate percentages of RPA+ cells. What was the treatment for the second panel?

Minor comments:

On page 3, there are 2 references in a different style than the rest of the manuscript and are missing from the list of references (Smits et al., 2006; Zou et al., 2002).

Page 5, sentence ending in "accumulate during mitosis and DNA replication" is missing a reference.

Many spelling and grammar mistakes throughout manuscript, authors should carefully go through and correct the text and figures. Few examples highlighted below.

Page 4 reads "the RPA-coated ssDNA can only active one round of ATR". This should be presumably read "activate one round of ATR".

Page 7 reads "ATR is recruited and active on RPA-coated ssDNA". Presumably this should read "recruited to and active on" or "recruited and activated".

Figure S4B, "breaks" and "metaphase" are misspelled on y-axes.

Could the authors include one or two introductory sentences on ATR at the start of the abstract?

Can the authors elaborate on the choice of mutation used to generate the AtrKD allele?

It is stated in the text that Atr^{+/KD} mice were of normal size, but this is not shown in Figure 1.

Throughout the figures, authors indicate significantly different results using stars. They however do not indicate the meaning behind the number of stars ($p < 0.05$). Furthermore, authors should include actual p-values in figures, figure legend, or in the text for both significant and non-significant differences.

Also, for several figures there is no indication of significance, even if differences appear to be significant by eye. Authors should evaluate each figure and add statistical significance where possible.

Reviewer #2 (Remarks to the Author):

This study describes the generation and characterizations of ATR^{+/KD} mice and derivative cells. In addition, the authors used FRAP to study the turnover of GFP-ATR and GFP-RPA at DNA damage sites. The authors did a good job in characterizing the various phenotypes of ATR^{-/+KD} mice and cells. Many of the phenotypes of ATR^{+/KD} mice and cells are consistent with partially defective ATR functions. Since many labs have used ATR-KD as a dominant negative mutant to inhibit ATR activity in cell based studies, it is perhaps not too surprising to see the in vivo and in vitro defects of ATR^{+/KD} mice and cells. The FRAP data in Fig. 3 and 5d are the most interesting results in this study. Although the observations in these FRAP experiments are quite interesting, the underlying mechanisms are not clearly worked out. The model that ATR regulates downstream repair by enhancing RPA exchange on ssDNA still needs more direct evidence. This study would be strengthened by additional biochemical studies that directly test the impacts of ATR activity on the behaviors of ATR and RPA on DNA.

1. Figs. 1 and 2 are focused on the meiotic and B-cell defects of ATR^{+/KD} mice. In some experiments, ATR^{+/KD} was clearly more defective than ATR^{+/-} (1d, 1f, 2a, 2b, 2d, 2e). In others, ATR^{+/KD} and ATR^{+/-} were similarly defective (2c). Can this discrepancy be explained?

2. In Fig. 3b, the delay of GFP-ATR recovery could be attributed to slow release of bleached GFP-ATR or slow recruitment of fresh GFP-ATR. The recruitment of fresh GFP-ATR during recovery may be different from the recruitment of GFP-ATR after the initial damage because many things may have changed at the damage sites after the initial response. Can the authors distinguish these two possibilities?

3. In Fig. 4 and 5a-c, the defects of ATR^{+/KD} cells in the replication stress response are very consistent with the known functions of ATR. However, these experiments don't seem to provide

new insights into the defects of ATR-KD. The increase of RPA could come from the replication problems caused by ATR-KD and not necessarily an effect of trapping.

4. In Fig. 5d, the effects of ATRi on GFP-RPA recovery seem subtle. As a comparison, the difference between ATR-WT and ATR-KD in Fig. S3c seems more obvious. The conclusions from these two figures are opposite: ATRi inhibits RPA exchange, but ATR-KD is not defective for localization. I am not convinced by these interpretations.

5. In Fig. 6, RAD51 foci formation and fork restart are defective in ATR+/KD cells, but these defects may not be caused by reduced RPA exchange on ssDNA. ATR could directly phosphorylate HR and fork proteins in these responses. The dominant negative effects of ATR-KD on ATR activity could explain these defects without involving RPA trapping into the model.

Reviewer #3 (Remarks to the Author):

This manuscript studies the effect of ATR inhibition in comparison to ATR loss and its consequences for checkpoint activation and DNA repair. To mechanistically study this, the authors created a mouse model expressing a kinase-dead ATR and analyzed its phenotype in comparison to mice lacking a wild-type copy of the *Atr* gene. The authors report that ATR kinase activity is required to for sex body formation in mouse spermatocytes and thus to promote male fertility. They also found that ATR function is required to promote normal development of lymphocyte. Using this mouse model and ATR inhibitors, the authors show that ATR is exchanged rapidly at DNA damaged sites and this depends on ATR-kinase activity and is required for the downstream activation of the ATR-dependent DDR pathway. ATR-KD cells accumulate RPA on the chromatin, part of it gets hyperphosphorylated and blocks lesion repair. According to the authors, this may explain the hypersensitivity of ATR-KD mutant cells to ssDNA-causing agents. Altogether, these results suggest mechanistic differences between ATR inhibition and ATR loss that have critical implications in order to understand the different behavior these different cell lines may have to certain treatments or conditions.

Overall, the paper is well written and the conclusions are supported by the presented data. ATR is a key regulator of chromosome integrity and the functional analysis of the kinase-dead and its comparison to the mutant expressing reduced levels of ATR protein will be of interest to a broad readership. Nonetheless, a few points should be addressed before this manuscript is considered for publication:

1.-In my opinion the description of the meiotic phenotype is very interesting but unfortunately, I think the authors did not describe it appropriately.

- The authors report that *Atr*+/*KD* mice are sterile due to a significant reduction in sperm counts and a meiotic arrest. But this arrest is not properly studied. Authors should specify at which spermatogenic stage the seminiferous tubules are arrested. At what stage are the arrested primary spermatocytes? Also, the provided images in Fig 1e are too small to judge the occurrence of this arrest. I would urge the authors to provide larger images or zoomed-in pictures of relevant parts of the tubules.
- Are these arrested spermatocytes eliminated by programmed cell death?
- Based on the images provided it seems *ATR*+/*KD* are able to synapse their homologs. This suggests that meiotic recombination is not affected by *KD* mutation. However, it would be good to report the evolution of meiotic recombination markers, like RPA, RAD51/DMC1, and MLH1 in *ATR*+/*KD* in comparison to *ATR*+/-.
- Based on the results found on the analysis of RAD51 foci formation on HU-treated *ATR*+/*KD* cells, one would expect that loading of RAD51 on the non-homologous portion of the sex chromosomes would be compromised in these mutant spermatocytes. Could the authors perform this analysis? Moreover, finding different RAD51 loading efficiencies on DSBs on the autosomes and on the sex chromosomes would validate their model in which ATR kinase activity is particularly important for those lesions that expose ssDNA fragments for longer periods of time.

- Furthermore, it would be good to provide an analysis of meiotic prophase progression to certify that this mutation does not interfere with homologous chromosome synapsis. This could be easily done by analyzing the proportion of spermatocytes found at the different stages of meiotic prophase (leptotene, zygotene, pachytene and diplotene) in control and mutant testis.
- 2.-Page 7. Mention the name and concentration of the ATR inhibitor used in this experiment.
- 3.-Page 9. What is the statistical analysis performed on the chromosome abnormalities study reported in the first paragraph of this page and on figure 4a and Fig S4b?. It seems to me that Atr+/- cells treated with aphidicolin behave differently than wild-type.
- 4.-What is the statistical analysis performed on photobleaching experiments?
- 5.-Page 10. Please provide more details to explain how does the flow cytometry-based method used to analyze the levels of chromatin-bound RPA work.
- 6.-Page 11. Please provide a reference for the following sentence "consistent with the hypothesis that only a subset of RPA coated ssDNA is occupied by ATR-ATRIP at any given time".
- 7.-Page 12. Based on the presence of RPA foci on the autosomes during meiotic prophase, I don't think it is true that meiotic recombination at autosomes involves transient ssDNA formation. RPA foci appear at leptotema (early meiotic prophase) and are present up to mid pachynema, thus lasting almost the entire length of meiotic recombination. Thus, I think the authors should elaborate a bit more about this and extend this section of the discussion with the new meiotic analysis that I suggested.
- 8.-Page 18. Please check the figures cited in the first section of the methods, I think it should be Fig S1, instead of Fig S2.
- 9.-Page 18. Please provide a brief description of the AtrC and Atr- alleles.

We thank all the reviewers for their time and effort in preparing constructive and thorough reviews of our manuscript. We appreciate the positive feedbacks that highlighted the novelty and importance of our work. We also appreciate the comments and detailed suggestions that have helped us to significantly improve our manuscript. We have revised the manuscript text, figures, and supplemental material based on reviewer feedbacks, and we believe that we have addressed each of the concerns that were raised during the review. Below, we have addressed each of the reviewer comments point-by-point, with the original reviewer comments in *italics and bold*, and our responses in regular font.

Reviewer #1:

The authors of this study generated a mouse model expressing a kinase dead form of ATR, and compare several phenotypes of these ATR+/KD mice to both WT and ATR+/- mice. They show that the ATR+/KD displays male infertility due to defects in spermatogenesis, as well as fragility at telomeres and rDNA loci. The authors link this phenotype to defects in RPA exchange on single stranded DNA, leading to lack of CHK1 phosphorylation. These results further elucidate the effects of ATR inhibition on genome stability, and as such this study should be of interest to the genome instability community.

We thank the reviewer for the positive assessment of our study.

-There are however some concerns that need to be addressed before this manuscript is suitable for publication, as outlined below.

-The authors state that Atr+/KD mice display reduced efficiency of class switch recombination (Figure 2C). This is only true compared to WT mice, but not compared to Atr+/- mice. This should be clearly stated in the text.

While there is a statistically significant reduction of IgG1⁺% between *Atr*^{+KD} (n=7) and *Atr*^{+/+} (n=3) B cells at day 3.5 (p=0.007**) and 4.5 (p=0.0464*) (Fig. 2C), the IgG1⁺% from *Atr*^{+/-} (n=3) B cells is not significantly different from either *Atr*^{+KD} or *Atr*^{+/+} B cells (p=0.616, p=0.6535 vs *Atr*^{+KD} for day 3.5 and 4.5 respectively, while p=0.0742, p=0.0768 vs *Atr*^{+/+} for day 3.5 and 4.5 respectively). We now describe these results explicitly in the main text and mark the p-values in Figure 2c and in the corresponding figure legend.

-In Figure 2D, the authors claim an increased telomere fragility in Atr+/KD cells. Are these differences significant? In either case, (lack of) significance should be clearly indicated in the figure and text.

Yes, there is a statistically significant increase (p=0.0104 vs *Atr*^{+/+} and p=0.0142 vs *Atr*^{+/-}) of telomere fragility in *Atr*^{+KD} B cells in comparison to *Atr*^{+/+} or *Atr*^{+/-} B cells. We now include the p-values in Figure 2d.

-The authors show increased fragility at telomeres and rDNA loci in Atr+/KD cells (Figure 2D and 2E), and subsequently claim that Atr+/KD cells show elevated expression of B-cell specific early replicating fragile sites (ERFS). It is unclear why the authors chose to assess expression levels of these genes associated with ERFS (not actual ERFS), instead of DNA breakage at ERFS as shown in

the original paper describing ERFs. Furthermore, HU appears to induce a relative 2-2.5x increased expression in Atr+/KD cells for both the ERFs genes and tubulin. Is this effect specific to ERFs?

We thank the reviewer to suggest the cytogenetic approach. We initially chose the ERFs but not the CFS, because those ERFs were identified and characterized in murine splenic B cells (activated with conditions very similar to ours) and ATR inhibition was found to induce instability at the ERFs (Barlow *et al*, *Cell*, 2013). Following the reviewer’s suggestion, we performed cytogenetic analyses at one of the best characterized ERFs – the GIMAP locus (shown in Figure 2f and below). While there is no evidence for breaks at or near the GIMAP locus in 184 metaphases from two independently derived *Atr*^{+/+} B cell cultures, we identified 17 aberrations including breaks, radials and fusions adjacent to GIMAP locus in 193 metaphases (8.8 per 100 metaphases) from *Atr*^{+/KD} B cells (all treated with mild HU, 0.25mM) (Fig. 2f). The same standard (either proximity or splitting) originally used to define ERFs after high dose HU (10mM) (Barlow *et al*, *Cell*, 2013) was used here to score aberration at the GIMAP locus. We have now presented the FISH data, rather than the expression data, in Figure 2f to support increased fragility in the *Atr*^{+/KD} cells.

	Total	GIMAP	
Genotype	metaphases	aberrations	
ATR+/+	127	0	  normal    ATR+/KD   
ATR+/+	57	0	
ATR+/KD	138	14	
ATR+/KD	55	3	

-Related to the previous point, if the authors can show increased breakage at ERFs, it would be interesting to see if this is also true for common fragile sites, since it is known that ATR regulates CFS breakage.

As the reviewer has pointed out, ATR has been implicated in fragile sites stability (Casper *et al*, *Cell*, 2002). We have now analyzed the expression of two common fragile sites (CFSs): Fra14A2 (FRA3B in humans) and Fra8E1 (FRA16D in humans), in activated B cells upon 0.5 μM Aphidicolin treatment (overnight, ~20 hours). Under this condition, we only detected increased expression in one out of the 4 pair of primers. This could be due to the dose or time used in our experimental system. Given this result and the potential different mechanisms by which fragile site instability might be induced by HU vs Aphidicolin, at ERFs vs CFS, we felt that these findings are not conclusive at the current stage. We did not include them in the manuscript, but provide here to the reviewer as preliminary data (see graph

below, a and b indicate two different pair of oligos covering the fragile site).

Atr+/+ and Atr+/KD activated B cells were treated with 0.5 μ M Aphidicolin for 20 hours at day 3.5 after stimulation and RNA was collected for RT-qPCR. RT-qPCR was performed on two fragile sites, Fra14A2 and Fra8E1, using two different pair of oligos (a and b). Relative expression was normalized to actin.

-They measure rDNA instability by assessing the number of rDNA clusters different from 8. Can they give a reference that states that the physiological number of clusters is 8? Did ATR^{+/KD} lead to greater than or less than 8 cluster or both? What is the interpretation of these findings? Are there breaks associated with rDNA in met(a)phase spreads?

The exact number of mouse rDNA clusters varies between different mouse strains. Due to the repetitive nature of the rDNA sequence (estimated to be 200~400 copies per cell) and the divergence between different strains of mice, the precise location and number of rDNA are not available in mm9 and mm10. In the study by Kurihara et al (Kurihara et al., 1994), the authors used both silver staining (sensitively mapping transcriptionally active rDNA) and in situ hybridization (mapping all rDNA, but potentially less sensitive) to map the chromosomal location of rDNA clusters in laboratory strains of mice and show that 129/J strain has 8 rDNA clusters on chromosome 12,13,14,15,16, 17, 18, 19 (with the most transcriptionally active clusters on chr 12,18,19 and 16 in splenic B cells). Our Atr mice are all in the 129 background. We have now cited this reference in our manuscript.

The murine rDNA FISH probe that we developed in house with the guidance of Dr. Brian McStay, an expert in mammalian rDNA biology, reproducibly detects 8 pair of signals in 129 strains (~97% of metaphases from Atr^{+/+} B cells) (left example). Consistent with the variable size of the cluster, the fluorescence signal intensity varies between the clusters (Figure 2e, and below). Given the clusters are only detectable by cytogenetic means, when there are enough repeats in a given cluster. It is entirely possible that that additional smaller clusters might be discovered with more sensitive technologies. We have now included more details about the murine rDNA probe in the main text and in the method section.

As show in Figure 2e (and here), more than 8% of $ATR^{+/KD}$ cells have less than 8 clusters (blue bar) and 3% have > 8 clusters (red bar). We reason that the fluctuation of rDNA cluster number reflects aberrant recombination within (e.g. deletion) or between different clusters due to genomic instability, which would in turn give rise to less (deletion or intra-cluster recombination) or more (splits/translocation or recombination with a small, previously undetectable cluster) clusters over time.

Breaks within a given genomic region leads to split of the FISH signal. Since there are 8 pair of rDNA clusters per cell, any split or translocation within a given cluster would give rise to “additional rDNA cluster”, therefore we cannot confidently identify “breaks” with the current approach. Moreover in murine cells, the rDNA clusters reside near the centromere. Unrepaired breaks (not joined to another locus) in this region would lead to the loss of the entire chromosome arm and likely compromise viability. Given these considerations, we did not score rDNA breaks in unchallenged $Atr^{+/KD}$ B cells.

-They should explain why photobleaching experiments not performed in $ATR^{+/KD}$ cells, but WT cells treated with ATRi.

The main reason is technical, since the endogenous ATR is not GFP/RFP tagged in the $Atr^{+/KD}$ MEFs for live cell imaging analyses. Transient transfection and expression of ectopic ATR was proven to be inefficient in MEFs (always requires co-transfection of ATR-IP). The U2OS cells used in the study stably express GFP-ATR. As additional support, we performed FRAP experiments in U2OS cells transiently transfected with RFP-ATR WT or RFP-ATR-KD (together with ATRIP, see Figure S4b and S4c) and obtained consistent results, in which the exchange of ATR-KD is lower than that of ATR-WT, even in the presence of WT-ATR (endogenous and stably expressed as GFP fusion).

-Figure 5C, show percentages of RPA+ cells in the left panel, and absolute values instead of fold increases in right panel. Please indicate if differences between genotypes are significant.

We now marked the actual percentage of the RPA+ cells in both the left and the right panels (with means \pm SD and p value) of Figure 5b and 5c. There is significant more chromatin bounded RPA in $Atr^{+/KD}$ cells than in both $Atr^{+/-}$ and $Atr^{+/+}$ cells in the two doses of HU (0.1 and 0.2mM) tested.

-Figure 6C, show ratio of IdU/CIdU fiber length instead of absolute IdU fiber length.

We thank the reviewer for this thoughtful suggestion regarding the contribution of basal fork progression speed among different Atr -deficient cells. We have now provided the graph (below) that shows the ratio of IdU/CIdU fiber. Even after “normalize” against the CIdU portion, the $Atr^{+/KD}$ cells show significantly less recovery after HU challenging than $Atr^{+/-}$ and $Atr^{+/+}$ cells. We would like to point out, in our experiments (diagramed above the Figure 6C and also below), the IdU was added after HU was washed away. In this case, the length of IdU fiber (1hr) reflects fork restart after HU induced stalling of ongoing forks (defined by positive CIdU incorporation, 20 min pulse). The length of the CIdU fiber might vary not only due to the fork speed, but also at what time during the 20 min pulse, the particular origin starts dNTP/CIdU incorporations. In similar experiments published before, the IdU length alone is often used to measure fork progression and/or recovery (e.g. Figure 4 and 6 in (Bass et al., 2016)). For this reason, and given the identical conclusion reached by the IdU alone or IdU/CIdU ratio, we chose to show the length of IdU fibers in Figure 6C, which show defective fork recovery in

Atr^{+/-} and *Atr*^{+/*KD*} cells and a clearly significant more severe defects in the *Atr*^{+/*KD*} cells. Again, we sincerely thank the reviewer for his/her thoughtful comments.

Figure S5B, indicate percentages of RPA+ cells. What was the treatment for the second panel?

Due to the addition of new data, this figure has been renamed Figure S6b. We have now marked the percentage of RPA+ cells on each panel (as shown below). The treatment of the second panel is HU only, without any specific kinase inhibitors. This information is also marked on the upper left corner of each panel now (Figure S6b) (also see the replicate below).

Minor comments:

-On page 3, there are 2 references in a different style than the rest of the manuscript and are missing from the list of references (*Smits et al., 2006; Zou et al., 2002*).

We have fixed this reference error.

-Page 5, sentence ending in “accumulate during mitosis and DNA replication” is missing a reference.

This sentence is now in page 7. We identified four representative references (*Chan et al., 2018; Krawczyk et al., 2014; Sfeir et al., 2009; Vader et al., 2011*). The Sfeir paper shows increased ssDNA at the fragile telomere. The Chan et al. paper reported nicks and ssDNA at the fragile sites. The Vader et.

al. and Krawczyk et al. papers reported increase ssDNA at rDNA in yeast and mammalian cells respectively.

-Many spelling and grammar mistakes throughout manuscript, authors should carefully go through and correct the text and figures. Few examples highlighted below.

We have further proof-read the manuscript with the help of native speakers.

-Page 4 reads “the RPA-coated ssDNA can only active one round of ATR”. This should be presumably read “activate one round of ATR”.

We have fixed this grammar error.

-Page 7 reads “ATR is recruited and active on RPA-coated ssDNA”. Presumably this should read “recruited to and active on” or “recruited and activated”.

We have fixed this grammar error.

-Figure S4B, “breaks” and “metaphase” are misspelled on y-axes.

This figure is now Figure 4A. We have fixed the spelling errors.

-Could the authors include one or two introductory sentences on ATR at the start of the abstract?

We now included an introductory sentence in the abstract. The abstract is limited for 150 words. We are sorry for the abrupt entry to the main findings.

-Can the authors elaborate on the choice of mutation used to generate the AtrKD allele?

We have now explained the rationale in the main text. Briefly, we chose the D2466A mutation, since this mutation successfully eliminates the kinase activity of ATR without affecting the protein stability in two orthologues of ATR - in yeast (Mec1, D2224A) (Mallory and Petes, 2000) and human (ATR, D2475A) (Barr et al., 2003). In recent structural studies, the corresponding D2224 of Mec1 has been identified as a critical ATP handling residue and is predicted to abrogate kinase activity without affecting the overall folding of Mec1 (Wang et al., 2017). Moreover the corresponding mutation in the related kinases (ATM and DNA-PKcs) leads to stable expression of the “kinase dead” proteins in the respective mouse models we previously generated (Jiang et al., 2015; Yamamoto et al., 2012).

-It is stated in the text that Atr+/KD mice were of normal size, but this is not shown in Figure 1.

We have now included the weight of the mice in Figure S1f. We apologize for this oversight.

-Throughout the figures, authors indicate significantly different results using stars. They however do not indicate the meaning behind the number of stars ($p < 0.05$?). Furthermore, authors should include actual p -values in figures, figure legend, or in the text for both significant and non-significant differences.

The definition of the star (asterisk) was previously listed in the method section (* $p \leq 0.05$, ** $p \leq 0.01$, *** $p \leq 0.001$, **** $p \leq 0.0001$). We now include the actual p values in the figures, wherever possible for both significant and non-significant pairs. Due to space limitations, when we are not able to mark the p value for all possible pairwise comparisons, we state the p value in the figure legends (such as Supplementary figure 2c and Supplementary figure 5a).

-Also, for several figures there is no indication of significance, even if differences appear to be significant by eye. Authors should evaluate each figure and add statistical significance where possible.

As stated above, we have now marked the p values directly above each figure, wherever possible. Due to space limitation, we could not mark the p value for each pair wise comparison and chose to focus on biologically important pairs and state the p value for the other pairs in the figure legend. For FRAP experiments, we now show the maximal recovery as bar graphs with the p values marked on the graph. In a few exceptions, when it would be too crowded to mark the p value directly in the figure, we include statistic statements in the figure legends (for example, see Figure 2c and Supplementary Fig. 2c).

Reviewer #2 (Remarks to the Author):

This study describes the generation and characterizations of ATR+/KD mice and derivative cells. In addition, the authors used FRAP to study the turnover of GFP-ATR and GFP-RPA at DNA damage sites. The authors did a good job in characterizing the various phenotypes of ATR-/KD mice and cells. Many of the phenotypes of ATR+/KD mice and cells are consistent with partially defective ATR functions. Since many labs have used ATR-KD as a dominant negative mutant to inhibit ATR activity in cell based studies, it is perhaps not too surprising to see the in vivo and in vitro defects of ATR+/KD mice and cells. The FRAP data in Fig. 3 and 5d are the most interesting results in this study. Although the observations in these FRAP experiments are quite interesting, the underlying mechanisms are not clearly worked out. The model that ATR regulates downstream repair by enhancing RPA exchange on ssDNA still needs more direct evidence. This study would be strengthened by additional biochemical studies that directly test the impacts of ATR activity on the behaviors of ATR and RPA on DNA.

We thank the reviewer to bring up the previous cell based studies. Indeed ectopic expression of ATR-KD in human cells with endogenous WT-ATR causes complete cell cycle arrest (Barr et al., 2003). In fact, given the strong “dominant negative” effects (cell lethal) of the ectopic expression of ATR-KD, it is “unexpected” that *Atr*^{+/*KD*} mice are viable, of normal size and even fertile (female). We have now discuss our *in vivo* finding in the context of previous cell based study in the discussion. We also note that the thorough characterization of the *Atr*^{+/*KD*} mice model uncovered novel and unique tissue and organs specific vulnerability to Atr-KD *in vivo*, including X-Y body function and lymphocytopenia, which were not possible from cell line based studies. These phenotypes pointed out potential tissue specific toxicities (i.e. immune suppression and fertility changes) for the ATR kinase inhibitors and can be directly tested during clinical development of ATR inhibitors. Mechanistically, we now also show that the “inter-origin” distance in the *Atr*^{+/*KD*} MEFs is much shorter than that of *Atr*^{+/*+*} and *Atr*^{+/*-*} MEFs (Figure 6d), suggesting the apparently “normal proliferation and development” of the *Atr*^{+/*KD*} cells and

mice are in part due to compensatory firing of dormant replication origins – owing to Atr-KD blockade, reduced ATR signaling and others. Those findings provide an explanation for why the apparent normal *Atr*^{+/*KD*} cells and mice, and by extension, ATR inhibitor treated patients and animal models, are vulnerable to additional genotoxic challenge. We now present those data in Figure 6d and discuss their implications in the discussion section.

We thank the reviewer for his positive feedback on the FRAP analyses. In an effort to provide additional biochemical evidences that support the impact of ATR-KD on the behavior of RPA on the DNA, we performed chromatin fractionation with or without HU. Consistent with the observation in FRAP and the flow cytometry analyses of chromatin bounded RPA, chromatin bounded RPA and its phosphorylated forms (T21 and S4S8) increase specifically in *Atr*^{+/*KD*} cells, not in control cells (*Atr*^{+/+} and *Atr*^{+/-} cells) (Figure 5d). These data further support that the presence of ATR-KD protein leads to retention of RPA on chromatin. Together with the FRAP data, which shows a direct effect of catalytic inactive ATR on the exchange of a subset of RPA. Our finding supports a model in which ATR kinase activity regulates the exchange of **ATR itself** both directly (through trapping) and indirectly (through reduced ATR activity) to increase chromatin retention of RPA and affect downstream repair. We apologize for any confusion we had created in the previous version. We now clarify these points in the discussion (last paragraph).

1. Figs. 1 and 2 are focused on the meiotic and B-cell defects of ATR+/*KD* mice. In some experiments, ATR+/*KD* was clearly more defective than ATR+/- (1d, 1f, 2a, 2b, 2d, 2e). In others, ATR+/*KD* and ATR+/- were similarly defective (2c). Can this discrepancy be explained?

We thank the reviewer to point out this difference between *in vivo* characterization and short term *in vitro* culture. *Atr*^{+/*KD*} mice are clearly more defective than *Atr*^{+/-} mice in all the “*in vivo*” analyses – including male meiosis, thymocytes number and recirculated B cells frequency. In contrast, during class switch recombination (Figure 2c) and *in vitro* proliferation assay (Figure S5c), while there is a significant difference between *Atr*^{+/*KD*} and *Atr*^{+/+} cells, the difference between *Atr*^{+/*KD*} and *Atr*^{+/-} cells does not reach statistical significance. Notably, *Atr*^{+/-} cells are also not significant different from *Atr*^{+/+} controls in these assays. Several possibilities might contribute to this “discrepancy”. Most importantly, class switch recombination and proliferation assay are “short-term” with only a few cell divisions (n<4 doubling). In contrast, the marked difference between *Atr*^{+/*KD*} and *Atr*^{+/-} mice in meiosis and lymphocyte development reflects the “accumulative” effects after many cell divisions. During lymphocyte development, each round of V(D)J recombination is followed by robust proliferation to increase the pool carrying particular rearrangements. Consistent with the accumulative effects, cytogenetic analyses reveal mild, yet consistent increase of genomic instability in untreated *Atr*^{+/*KD*} cells, which could only lead to significant difference in cell numbers and development outcomes when they are allowed to accumulate exponentially through cell proliferation. In an oversimplified example, if the replication efficiency is 95% in *Atr*^{+/*KD*} cells and 98% in *Atr*^{+/-} cells, the difference in two cell division is $0.95^2=0.9025$ vs $0.98^2=0.9604$ (less than 6%, might be masked by experimental errors). But after 10 cell divisions, the difference would be $(0.95^{10} = 0.5987$ vs $0.98^{10}=0.8170$, more than 20%). We believed that this explains the substantial difference between *Atr*^{+/-} and *Atr*^{+/*KD*} mice *in vivo*, vs the relatively mild or no difference between them in short term culture experiments, highlighting the power of animal model studies. We have now modified the main text to emphasize this accumulative effect. We sincerely thank the reviewer to bring this point up.

2. In Fig. 3b, the delay of GFP-ATR recovery could be attributed to slow release of bleached GFP-ATR or slow recruitment of fresh GFP-ATR. The recruitment of fresh GFP-ATR during recovery may be different from the recruitment of GFP-ATR after the initial damage because many things may have changed at the damage sites after the initial response. Can the authors distinguish these two possibilities?

Yes. We thank the reviewer to bring up this valid concern. To distinguish these two possibilities, we reanalyzed several sets of FRAP experiments, in which we transfected either RFP-ATR-WT or RFP-ATR-KD (both with Flag-tagged ATRIP) into U2OS cells with stable expression of GFP-ATR-WT (Supplementary Figure S4d). Consistent with our main findings, the recovery of RFP-ATR-KD is more limited than that of RFP-ATR-WT, despite of the presence of the GFP-ATR-WT and the endogenous WT-ATR. Moreover, consistent with the “dominant negative effect” of ATR-KD, co-expression of the RFP-ATR-KD attenuates the recovery of the GFP-WT-ATR in the same cells. Most importantly, there is no significant difference on the recovery between RFP-ATR-KD and GFP-ATR-WT expressed in the same cells (the orange bars vs the dark green bars), suggesting the reduced recovery is NOT due to different recruitment kinetics of ATR-WT vs ATR-KD, but likely due to slow release of the bleached ATR molecules on DNA, which prevent the recovery of both WT and ATR-KD in the same cells. We now added this data and the discussion in the manuscript (Supplementary Figure S4d and also replicated here).

3. In Fig. 4 and 5a-c, the defects of ATR^{+ /KD} cells in the replication stress response are very consistent with the known functions of ATR. However, these experiments don't seem to provide new insights into the defects of ATR-KD. The increase of RPA could come from the replication problems caused by ATR-KD and not necessarily an effect of trapping.

First of all, we would like to emphasize that the data in Fig 4 and 5 a-c show clear differences between genetically defined *Atr*^{+ /KD} and the *Atr*^{+ /-} cells that can only be attributed to the presence of ATR-KD protein, not generic lack of ATR. Second, the FRAP experiments in Figure 5f and 5g, reveal measurable (10%), consistent and significant (p<0.05) delay of RPA exchange at DNA damage sites in ATR inhibitor treated cells, but not in CHK1 inhibitor treated cells. The time frame of those experiments (within 1hr about ATR addition, and within 15 min of initial damage) and the lack of effect after CHK1 inhibition, strongly suggest that the reduced RPA exchange is caused by direct and local effect of the

catalytic inactive ATR, not indirect effect in reducing ATR-CHK1 signaling. Although it is currently impossible to know how much of the chromatin retention of RPA is caused by compromised ATR signal in ATR-KD expressing cells and how much is caused by direct trapping of RPA by ATR-KD, the above evidence clearly support direct trapping of RPA as an important component and an unique consequence of ATR-KD (not just loss of ½ ATR, as the same phenotype is NOT found in *Atr*^{+/-} cells).

4. In Fig. 5d, the effects of ATRi on GFP-RPA recovery seem subtle. As a comparison, the difference between ATR-WT and ATR-KD in Fig. S3c seems more obvious. The conclusions from these two figures are opposite: ATRi inhibits RPA exchange, but ATR-KD is not defective for localization. I am not convinced by these interpretations.

Fig5d (now Figure 5e, 5f, 5g) measures RPA exchange and FigS3C (now S4C) measures ATR and ATR-KD exchange in the presence of endogenous WT-ATR. The findings from the two figures are both consistent with our working model and are NOT opposite to each other.

We apologize for the confusion in the figure legend for Fig. S3b and S3c (now S4b and S4c), which together with the small font of the axis title, might cause the reviewer to misunderstand Figure S3C (now S4c) as the recruitment kinase (rather than FRAP) of ATR vs ATR-KD. We now separated the figure legends. Briefly, the representative images in Figure S4b show the recruitment of ATR and ATR-KD (both tagged with RFP) are similar. This result is also supported by Fig S4a, in which ATR kinase inhibitor does not affect ATR recruitment (Fig S4a). Based on the similar recruitment kinetics, we then performed the FRAP experiment, which is quantified in Figure S4c - orange line for RFP-ATR-KD and red line for RFP-ATR-WT (both with ATR-IP and in U2OS cells with endogenous WT-ATR). The data suggest that ATR-KD exchange is less efficient than ATR-WT, even in the presence of endogenous WT-ATR, consistent with the dominant role of ATR-KD and in line with the similar result obtained using ATRi (VE-821) in Figure 3b.

On the other hand, Figure 5d (now 5e, 5f and 5g) measured the recovery of GFP-RPA itself. We agree and also clearly stated in the text, the impact of ATR inhibition on RPA exchange is “moderate (10%) yet consistent and significant ($p < 0.05$)”. This result is highly reproducible and is consistent with the fact that RPA coated ssDNA also recruits RAD51, RAD52 and others in addition to ATR-ATRIP.

Finally, partially delay of RPA exchange on focal DNA damage caused by ATR inhibition does not necessarily leads to defects in ATR-KD recruitment or localization defects. In vivo, chromatin bound RPA could be lifted by competitive binding of free RPA (Ma et al., 2017), or ultimately by on-going lagging strand synthesis (during DNA replication), RAD51 loading (during HR) and DMC1 loading (during meiosis). The intensity of ATR foci at DNA damage sites reflects a combination of recruitment, repair, duration of stay and potentially other factors, and is not solely determined by RPA exchange, especially not the frequent and rapid exchange ($t_{1/2}$ with 7 seconds) we detected here.

5. In Fig. 6, RAD51 foci formation and fork restart are defective in ATR+/KD cells, but these defects may not be caused by reduced RPA exchange on ssDNA. ATR could directly phosphorylate HR and fork proteins in these responses. The dominant negative effects of ATR-KD on ATR activity could explain these defects without involving RPA trapping into the model.

We apologize for the complexity of our writing and the model, which might lead to the misunderstanding of our model. As stated for the response to comment 3 from this reviewer, we agree with the reviewer that the “moderate” direct trapping of RPA itself, although significant and important, is not sufficient to explain all the phenotype of *Atr*^{+/*KD*} mice and the reduced RAD51 foci. We now clarify the model as following.

Our model states that the reduced exchange of **ATR itself** and RPA work **together** to limit the RAD51 foci formation. In the revisions, we have further emphasize the combined effects (in the results and also in the discussions). It is worth to mention that we observed reproducible and statistically significant defects in RPA exchange via FRAP and only in ATR inhibitor treated, but not in CHK1 inhibitor treated cells, which clearly indicates the existence of a “direct and local” impact of the catalytic inactive ATR on RPA exchange, beyond what might be expected from reduced ATR signaling in general. Moreover the unique physiological phenotypes of *Atr*^{+/*KD*} mice over *Atr*^{+/-} models in meiosis and lymphocyte development, unequivocally support a structural role of ATR-KD that cannot be simply explained by the lack of ATR activity alone. Thus, although it is not possible to precisely determine how much of the *Atr*^{+/*KD*} phenotype, including reduced RAD51 foci formation, is caused by defective ATR exchange itself and how much is caused by the subsequent reduction in RPA exchange, the overwhelming evidence presented in this study support our model in which reduced exchange of ATR itself and RPA together contribute to the overall phenotype in *Atr*^{+/*KD*} cells.

Reviewer #3 (Remarks to the Author):

This manuscript studies the effect of ATR inhibition in comparison to ATR loss and its consequences for checkpoint activation and DNA repair. To mechanistically study this, the authors created a mouse model expressing a kinase-dead ATR and analyzed its phenotype in comparison to mice lacking a wild-type copy of the Atr gene. The authors report that ATR kinase activity is required for sex body formation in mouse spermatocytes and thus to promote male fertility. They also found that ATR function is required to promote normal development of lymphocyte. Using this mouse model and ATR inhibitors, the authors show that ATR is exchanged rapidly at DNA damaged sites and this depends on ATR-kinase activity and is required for the downstream activation of the ATR-dependent DDR pathway. ATR-KD cells accumulate RPA on the chromatin, part of it gets hyperphosphorylated and blocks lesion repair. According to the authors, this may explain the hypersensitivity of ATR-KD mutant cells to ssDNA-causing agents. Altogether, these results suggest mechanistic differences between ATR inhibition and ATR loss that have critical implications in order to understand the different behavior these different cell lines may have to certain treatments or conditions. Overall, the paper is well written and the conclusions are supported by the presented data. ATR is a key regulator of chromosome integrity and the functional analysis of the kinase-dead and its comparison to the mutant expressing reduced levels of ATR protein will be of interest to a broad readership. Nonetheless, a few points should be addressed before this manuscript is considered for publication:

We thank the reviewer for the positive feedback on our study.

1.-In my opinion the description of the meiotic phenotype is very interesting but unfortunately, I think the authors did not describe it appropriately.

• The authors report that *Atr*^{+/*KD*} mice are sterile due to a significant reduction in sperm counts and

a meiotic arrest. But this arrest is not properly studied. Authors should specify at which spermatogenic stage the seminiferous tubules are arrested. At what stage are the arrested primary spermatocytes? Also, the provided images in Fig 1e are too small to judge the occurrence of this arrest. I would urge the authors to provide larger images or zoomed-in pictures of relevant parts of the tubules.

We thank the reviewer for highlighting the importance of the meiosis phenotype. Promoted by the suggestion, we moved some other panels in Figure 1 to supplementary, which provide space to show high resolution images of the developing testes (Figure 1d). We also determined at which stage the spermatogenesis is arrested in $Atr^{+/KD}$ mice by 1) performing TUNEL staining in developing testes (Figure 1d and Supplementary Figure 2a), 2) examining the distribution of spermatocytes at different stages (leptotene, zygotene, pachytene and diplotene) during meiosis prophase I from 3 week $Atr^{+/KD}$ and control mice (supplementary Figure 2c and 2d).

The TUNEL positive cells are mostly large primary spermatocytes (Figure 1d), suggesting defects in meiosis I. The stage analyses (supplementary Figure 2b and 2d) show a marked reduction of diplotene and a sharp decrease of diplotene vs pachytene ratio in $Atr^{+/KD}$ mice, indicating a developmental block between pachytene and diplotene transition. This is the stage when X-Y body is formed and MSCI occurs. Together these new data narrow the spermatogenesis defects in $Atr^{+/KD}$ mice to pachytene and diplotene transition, when X-Y body formation and MSCI occurs.

• Are these arrested spermatocytes eliminated by programmed cell death?

Yes. We performed TUNEL staining of testes from age matched $Atr^{+/KD}$, $Atr^{+/+}$ and $Atr^{+/-}$ mice. The analyses uncovered cluster of apoptotic cells (Figure 1d) and significant increase frequency of seminiferous tubules with cluster of apoptotic cells (>3 TUNEL+ cells per tube) in the testes of $Atr^{+/KD}$ mice (Supplementary Figure 2a).

• Based on the images provided it seems $ATR^{+/KD}$ are able to synapse their homologs. This suggests that meiotic recombination is not affected by KD mutation. However, it would be good to report the evolution of meiotic recombination markers, like RPA, RAD51/DMC1, and MLH1 in $ATR^{+/KD}$ in comparison to $ATR^{+/-}$.

In the revision, we have analyzed RAD51 and MLH1 foci in the spermatocyte spreads from $Atr^{+/KD}$, $Atr^{+/+}$ and $Atr^{+/-}$ mice. The appearance and density of RAD51 foci (in leptotene) – an indicator for early meiotic recombination, and the number of MLH1 foci per cells (in pachytene) – a marker for meiotic crossover are not significant different among $Atr^{+/KD}$, $Atr^{+/+}$ and $Atr^{+/-}$ mice (supplementary figure 2f and 2g). Together with the normal fertility of the female $Atr^{+/KD}$ mice (litter size information is now included in supplementary Figure 1h) and the normal synapse of the autosome in $Atr^{+/KD}$ spermatocytes (Fig 1e), these findings support normal autosome meiosis recombination and selective defects in X-Y body in $Atr^{+/KD}$ mice.

• Based on the results found on the analysis of RAD51 foci formation on HU-treated $ATR^{+/KD}$ cells, one would expect that loading of RAD51 on the non-homologous portion of the sex chromosomes would be compromised in these mutant spermatocytes. Could the authors perform this analysis? Moreover, finding different RAD51 loading efficiencies on DSBs on the autosomes and on the sex

chromosomes would validate their model in which ATR kinase activity is particularly important for those lesions that expose ssDNA fragments for longer periods of time.

We thank the reviewer for this thoughtful suggestion. We now analyzed the RAD51 foci in the spermatocyte spreads from *Atr*^{+/*KD*}, *Atr*^{+/+} and *Atr*^{+/-} mice (supplementary figure 2f). The results do not show overwhelming defects in RAD51 foci formation in *Atr*^{+/*KD*} spermatocytes. The majority of the RAD51 foci in Leptonene are in autosome. The normal appearance of RAD51 is consistent with the normal autosomes synapses in *Atr*^{+/*KD*} mice. Many differences between the HU induced Rad51 foci formation in MEFs and the physiological RAD51 foci during meiotic recombination, including the contribution of DMC1 and the fate of the recombination intermediates, could potentially explain this difference. Although interesting, this result is not very informative at this stage.

• Furthermore, it would be good to provide an analysis of meiotic prophase progression to certify that this mutation does not interfere with homologous chromosome synapsis. This could be easily done by analyzing the proportion of spermatocytes found at the different stages of meiotic prophase (leptotene, zygotene, pachytene and diplotene) in control and mutant testis.

We thank the reviewer for this constructive suggestions. As stated in the response to the first specific suggestion, we have analyzed meiotic prophase I progression from 3 week old *Atr*^{+/*KD*} and control mice. The results show a clear block at the pachytene to diplotene transition, consistent with defects in X-Y body formation and MSC1 (supplementary figure 2c and 2d).

2.-Page 7. Mention the name and concentration of the ATR inhibitor used in this experiment.

ATR inhibitor VE-821 was used at 10μM and CHK1 inhibitor LY2603618 was also used at 10μM. We have now specified the name and dose in the text (now page 9) and in the Figure legends.

3.-Page 9. What is the statistical analysis performed on the chromosome abnormalities study reported in the first paragraph of this page and on figure 4a and Fig S4b?. It seems to me that *Atr*^{+/-} cells treated with aphidicolin behave differently than wild-type.

The statistical analyses used is student t-test. We apologized for the appearance of many different error bars that crowded the figure. In the revision, we have simplified the figures, which show the p values (above the bar graph in Figure 4a and in the figure legend in Figure S5a), between the “overall frequency of metaphase with abnormalities – including breaks, fusion and mitotic catastrophe” (now Figure S5a, corresponding to the old Figure 4a) and “total number of breaks – including both chromatid, chromosomal and breaks resulted fusion” (now Figure 4a, corresponding to the old Figure S4b). We divided the bar by the mean frequency of each categories (color coded). Indeed, under Aphidicolin treated condition, *Atr*^{+/-} cells have significant more instability than wild-type cells, but significant less instability than the *Atr*^{+/*KD*} cells. We now stated this explicitly in the corresponding figure legend (now Figure S5a).

4.-What is the statistical analysis performed on photobleaching experiments?

For all the FRAP experiments, we calculate the $t_{1/2}$ and the maximal recovery on each cells using the previously described method (Harrington et al., 2002; Ostlund et al., 2006). For every FRAP experiment shown (Figure 3b, 3c, 5e-g, S4c, S6c) we have included bar graphs, showing the mean \pm SEM of the maximal recovery. Unpaired two-tailed t test was used to calculate the p value of significance between all the pairs analyzed and all the p values are reported above the bar graphs. Furthermore, we have included more details about the statistical analysis in the method section.

5.-Page 10. Please provide more details to explain how does the flow cytometry-based method used to analyze the levels of chromatin-bound RPA work.

This method was detailed in a Nature Protocol paper (Forment and Jackson, 2015) and have been used broadly to study chromatin RPA levels (for example in the Extended Data Fig. 9 in a recent Nature paper (Schrank et al., 2018)). In addition to the reference, we also briefly described the procedure in the method section as following.

“The recruitment of RPA to chromatin was performed essentially as described(Forment and Jackson, 2015) with minor modifications. This protocol allows to quantify only the fraction of RPA that is associated to chromatin and to measure in parallel the cell cycle distribution using Propidium Iodide (PI) staining. Briefly, untreated or HU-treated cells were permeabilized with PBS/Triton-X 0.2% for 5 minutes on ice (non-chromatin bound protein extraction step), washed with BSA 5%/PBS and then fixed with Paraformaldehyde 4% for 10 minutes at room temperature. Cells were then resuspended in BSA 5%/PBS for 1 hour, incubated with primary RPA32 antibody (NA19L, Calbiochem) for 2 hours at room temperature and then with Alexa Fluor 488-conjugated secondary goat anti-mouse antibody (Thermo Fisher Scientific) for 1 hour in the dark. Finally, DNA was stained with PI (Sigma) for 20 minutes at room temperature in the dark. More than 25,000 cells for every sample were acquired using CellQuest software on FACSCalibur instrument and analyzed using FlowJo.”

6.-Page 11. Please provide a reference for the following sentence “consistent with the hypothesis that only a subset of RPA coated ssDNA is occupied by ATR-ATRIP at any given time”.

This sentence is now at page 13. We have included several references show that ssDNA-RPA is not only bound by ATR-ATRIP, but it is a platform for other proteins, like the recombination factors Rad51, RAD52 and SMARCAL1 etc (Ma et al., 2017; Sugiyama and Kowalczykowski, 2002)

7.-Page 12. Based on the presence of RPA foci on the autosomes during meiotic prophase, I don't think it is true that meiotic recombination at autosomes involves transient ssDNA formation. RPA foci appear at leptonema (early meiotic prophase) and are present up to mid pachynema, thus lasting almost the entire length of meiotic recombination. Thus, I think the authors should elaborate a bit more about this and extend this section of the discussion with the new meiotic analysis that I suggested.

We thank the reviewer to point out this confusion. By “transient”, we did not mean the overall duration of RPA+ phase are short during meiosis, but rather try to imply each segment of RPA coated chromatin might be short live due to rapid replacement by RAD51/DMC1 and the subsequent recombination. Similarly, we also proposed that each RPA coated lagging strand segment is short lived and rapidly

replaced by ongoing lagging strand synthesis, although the replicating cells might well be positive for RPA foci throughout the entire S phase. We have now clarify this in the discussion (as below). We thank the reviewer for this important comment.

“Meiosis recombination at autosomes involving “transient” ssDNA formation that is quickly replaced by RAD51/DMC1, is largely normal in *Atr*^{+/*KD*} mice, but the resolution of X-Y bodies where persistent ssDNA accumulates at the non-homologous regions is abrogated (Fedoriw et al., 2015). Similarly, proliferation and S phase progression involving “transient” ssDNA at the lagging strands that are promptly replaced by ongoing lagging strand synthesis, are well tolerated in *Atr*^{+/*KD*} cells, but genomic instabilities are observed at telomeres, rDNA repeats and fragile sites, where ssDNA intermediates might accumulate during DNA replication and repair. Notably, we use the word “transient” to describe the short-live nature of each RPA-ssDNA segment. Meiotic cells or replicating cells can certainly have ssDNA and RPA coated ssDNA at different locations through meiosis or during the entire S phase.”

8.-Page 18. Please check the figures cited in the first section of the methods, I think it should be Fig S1, instead of Fig S2.

We have fixed this error. We thank the reviewer for pointing this out to us.

9.-Page 18. Please provide a brief description of the *Atr*^C and *Atr*⁻ alleles.

We now included a brief description in the materials and methods under the subtitle “ATR alleles used in the study”.

The *Atr*^C and the *Atr*⁻ alleles were generated by Drs. Eric Brown and David Baltimore and have been carefully characterized in a number of publications (Brown and Baltimore, 2003; Chanoux et al., 2009; Lee et al., 2012; Royo et al., 2013; Ruzankina et al., 2007). Briefly, the *Atr*^C allele contains two loxP sites flanking the two exons encoding the core kinase domain of ATR and the *Atr*⁻ allele contains the deletion for the first 3 exons, including the initiating methionine code of murine ATR (Brown and Baltimore, 2003).

Reference for the point by point responses

- Barr, S.M., Leung, C.G., Chang, E.E., and Cimprich, K.A. (2003). ATR kinase activity regulates the intranuclear translocation of ATR and RPA following ionizing radiation. *Curr Biol* *13*, 1047-1051.
- Bass, T.E., Luzwick, J.W., Kavanaugh, G., Carroll, C., Dugrawala, H., Glick, G.G., Feldkamp, M.D., Putney, R., Chazin, W.J., and Cortez, D. (2016). ETAA1 acts at stalled replication forks to maintain genome integrity. *Nat Cell Biol* *18*, 1185-1195.
- Brown, E.J., and Baltimore, D. (2003). Essential and dispensable roles of ATR in cell cycle arrest and genome maintenance. *Genes Dev* *17*, 615-628.
- Chan, Y.W., Fugger, K., and West, S.C. (2018). Unresolved recombination intermediates lead to ultra-fine anaphase bridges, chromosome breaks and aberrations. *Nat Cell Biol* *20*, 92-103.
- Chanoux, R.A., Yin, B., Urtishak, K.A., Asare, A., Bassing, C.H., and Brown, E.J. (2009). ATR and H2AX cooperate in maintaining genome stability under replication stress. *J Biol Chem* *284*, 5994-6003.

Fedoriw, A.M., Menon, D., Kim, Y., Mu, W., and Magnuson, T. (2015). Key mediators of somatic ATR signaling localize to unpaired chromosomes in spermatocytes. *Development* *142*, 2972-2980.

Forment, J.V., and Jackson, S.P. (2015). A flow cytometry-based method to simplify the analysis and quantification of protein association to chromatin in mammalian cells. *Nat Protoc* *10*, 1297-1307.

Harrington, K.S., Javed, A., Drissi, H., McNeil, S., Lian, J.B., Stein, J.L., Van Wijnen, A.J., Wang, Y.L., and Stein, G.S. (2002). Transcription factors RUNX1/AML1 and RUNX2/Cbfa1 dynamically associate with stationary subnuclear domains. *J Cell Sci* *115*, 4167-4176.

Jiang, W., Crowe, J.L., Liu, X., Nakajima, S., Wang, Y., Li, C., Lee, B.J., Dubois, R.L., Liu, C., Yu, X., *et al.* (2015). Differential phosphorylation of DNA-PKcs regulates the interplay between end-processing and end-ligation during nonhomologous end-joining. *Mol Cell* *58*, 172-185.

Krawczyk, C., Dion, V., Schar, P., and Fritsch, O. (2014). Reversible Top1 cleavage complexes are stabilized strand-specifically at the ribosomal replication fork barrier and contribute to ribosomal DNA stability. *Nucleic Acids Res* *42*, 4985-4995.

Kurihara, Y., Suh, D.S., Suzuki, H., and Moriwaki, K. (1994). Chromosomal locations of Ag-NORs and clusters of ribosomal DNA in laboratory strains of mice. *Mamm Genome* *5*, 225-228.

Lee, Y., Shull, E.R., Frappart, P.O., Katyal, S., Enriquez-Rios, V., Zhao, J., Russell, H.R., Brown, E.J., and McKinnon, P.J. (2012). ATR maintains select progenitors during nervous system development. *EMBO J* *31*, 1177-1189.

Ma, C.J., Gibb, B., Kwon, Y., Sung, P., and Greene, E.C. (2017). Protein dynamics of human RPA and RAD51 on ssDNA during assembly and disassembly of the RAD51 filament. *Nucleic Acids Res* *45*, 749-761.

Mallory, J.C., and Petes, T.D. (2000). Protein kinase activity of Tel1p and Mec1p, two *Saccharomyces cerevisiae* proteins related to the human ATM protein kinase. *Proc Natl Acad Sci U S A* *97*, 13749-13754.

Ostlund, C., Sullivan, T., Stewart, C.L., and Worman, H.J. (2006). Dependence of diffusional mobility of integral inner nuclear membrane proteins on A-type lamins. *Biochemistry* *45*, 1374-1382.

Royo, H., Prosser, H., Ruzankina, Y., Mahadevaiah, S.K., Cloutier, J.M., Baumann, M., Fukuda, T., Hoog, C., Toth, A., de Rooij, D.G., *et al.* (2013). ATR acts stage specifically to regulate multiple aspects of mammalian meiotic silencing. *Genes Dev* *27*, 1484-1494.

Ruzankina, Y., Pinzon-Guzman, C., Asare, A., Ong, T., Pontano, L., Cotsarelis, G., Zediak, V.P., Velez, M., Bhandoola, A., and Brown, E.J. (2007). Deletion of the developmentally essential gene ATR in adult mice leads to age-related phenotypes and stem cell loss. *Cell Stem Cell* *1*, 113-126.

Schrank, B.R., Aparicio, T., Li, Y., Chang, W., Chait, B.T., Gundersen, G.G., Gottesman, M.E., and Gautier, J. (2018). Nuclear ARP2/3 drives DNA break clustering for homology-directed repair. *Nature* *559*, 61-66.

Sfeir, A., Kosiyatrakul, S.T., Hockemeyer, D., MacRae, S.L., Karlseder, J., Schildkraut, C.L., and de Lange, T. (2009). Mammalian telomeres resemble fragile sites and require TRF1 for efficient replication. *Cell* *138*, 90-103.

Sugiyama, T., and Kowalczykowski, S.C. (2002). Rad52 protein associates with replication protein A (RPA)-single-stranded DNA to accelerate Rad51-mediated displacement of RPA and presynaptic complex formation. *J Biol Chem* *277*, 31663-31672.

Vader, G., Blitzblau, H.G., Tame, M.A., Falk, J.E., Curtin, L., and Hochwagen, A. (2011). Protection of repetitive DNA borders from self-induced meiotic instability. *Nature* *477*, 115-119.

Wang, X., Ran, T., Zhang, X., Xin, J., Zhang, Z., Wu, T., Wang, W., and Cai, G. (2017). 3.9 A structure of the yeast Mec1-Ddc2 complex, a homolog of human ATR-ATRIP. *Science* *358*, 1206-1209.

Yamamoto, K., Wang, Y., Jiang, W., Liu, X., Dubois, R.L., Lin, C.S., Ludwig, T., Bakkenist, C.J., and Zha, S. (2012). Kinase-dead ATM protein causes genomic instability and early embryonic lethality in mice. *J Cell Biol* 198, 305-313.

Reviewers' comments:

Reviewer #1 (Remarks to the Author):

This is an excellent revision and manuscript. I only have one remaining question. The defect in RAD51 foci due to defective RPA exchange after HU is interesting. Is this specific for ssDNA induced by HU, or is there also a defect in RAD51 loading after IR, which is presumably after RPA loading at resected DSBs?

Reviewer #2 (Remarks to the Author):

I appreciate the authors' response but still feel that the following issues need further attention. The lack of biochemistry is still a problem. The increase of RPA and phospho-RPA one chromatin in ATR+/KD cells is not necessarily a result of reduced RPA exchange - it may be a consequence of increased origin firing and induction of DNA damage. The functional contributions of ATR and RPA exchanges are still difficult to estimate. ATR-KD inhibits phosphorylation of many ATR substrates in addition to affecting ATR and RPA exchanges. It remains possible that the phenotypes of ATR+/KD cells are primarily caused by the reduced phosphorylation of ATR substrates.

Reviewer #3 (Remarks to the Author):

In this revised and improved version of the manuscript, the authors have addressed most of my requests. However, the authors have failed to show convincing results about proper RAD51 loading on mutant spermatocytes. The images provided in the Figure S2f are uninformative since no clear foci can be observed at 100% (or even at 400%) magnification. Also, based on the proper achievement of homologous chromosome synapsis displayed by the Atr+/KD spermatocytes, I would not expect differences that could be easily seen by the eye. Furthermore, the authors state that "the [...] frequency of RAD51 foci [...] were not significantly altered in the Atr+/KD spermatocytes", but they don't provide any data about the frequency of RAD51 foci found in controls or mutant spermatocytes. Thus, I think it is crucial to count the number of RAD51 foci found at leptotene stage in wt, hets, and Atr+/KD to complete this analysis. Apart from this one, I do not have any other comment on the manuscript, which I am sure it will be of interest to the broad readership of Nature Communications.

We thank all the reviewers for taking time to evaluate our revised manuscript and for their valuable insights, which had helped us further improve our manuscript. We are glad to see that all three reviewers appreciated our sincere efforts in the revision and described our manuscript as “excellent” and “sure to be of interest to the broad readership of Nature Communications”. In this 2nd revision, we have addressed all the remaining concerns 1) by measuring IR induced RAD51 foci (suggested by reviewer 1), 2) by quantify RAD51 foci in leptotene stage (suggested by reviewer 3) and 3) by further clarifying our model in the context of existing knowledge about ATR and ATR-KD and discussing the limitations and future directions for our study.

Original comments from the reviewers are ***bold and italicized***. All actual text changes in the manuscript are **highlighted** (yellow). For clarity, we did not track section-order changes or shift of figure numbers, when the text itself is not modified. A summary is provided at the end of this response to detail all figure numbers shifts caused by insertion of new panels.

Reviewer #1 (Remarks to the Author):

This is an excellent revision and manuscript. I only have one remaining question. The defect in RAD51 foci due to defective RPA exchange after HU is interesting. Is this specific for ssDNA induced by HU, or is there also a defect in RAD51 loading after IR, which is presumably after RPA loading at resected DSBs?

We thank the reviewer for the very positive statement about our manuscript and appreciate his/her thoughtful suggestions. We have measured IR (5Gy) induced RAD51 foci from *Atr*^{+/+}, *Atr*^{+/-} and *Atr*^{+/*KD*} MEFs in two independent experiments (with over 1000 individual cells counted per genotype). As shown in **Supplementary Fig. 6d** (also on the right), the frequency of cells with >5 Rad51 foci at 10hr after IR does not differentiate significantly among *Atr*^{+/+}, *Atr*^{+/-} and *Atr*^{+/*KD*} MEFs. This is different from the much reduced RAD51 foci in HU (2mM, 6hr) treated *Atr*^{+/*KD*} MEFs (Figure 6b), indicating the RAD51 loading defects in *Atr*^{+/*KD*} cells is specific for HU induced damage. Several differences between IR vs HU induced breaks might explain this selectivity.

First, HU uncouples DNA replication and directly induces extensive ssDNA and robust ATR activation. In contrast, IR primarily generates DNA double strand breaks, which activates ATM and DNA-PK kinases. As shown in the following Western blotting, IR induces robust phosphorylation of KAP1 and H2AX- two substrates of ATM kinase. Meanwhile IR induces nearly no detectable phosphorylation of CHK1 – a marker for ATR activation and minimal RPA1 phosphorylation (T21, as well as S4/8) – a sign of ssDNA formation. Correspondingly ATR inhibition and expression of ATR-KD have no

measurable impact on the level of IR induced RPA T21 or S4/8 phosphorylation, which is different from the major increase of HU induced RPA phosphorylation in *Atr^{+/-KD}* cells. So the limited ATR activation by IR explains the selective impact of ATR-KD on HU induced –RAD51 foci, but not on IR induced RAD51 foci.

[REDACTED]

Second, several studies suggest that the vast majority (~90%) of the IR induced DNA breaks are rapidly repaired (within 6hrs) by non-homologous end-joining (NHEJ) without significant DNA end-resection and, by extension, RPA loading (Riballo et al., 2004; Shibata et al., 2013). Indeed, while HU induces prominent accumulation of cells with > 5 RAD51 foci and corresponding reduction of cells with no detectable RAD51 foci in *Atr^{+/+}* MEFs, IR only induces a small shift of cells with more than 5 RAD51 foci beyond the baseline untreated conditions. Since a fraction of normal S phase cells might also have >5 RAD51 foci at a time (See panel on the left).

Third, due to the need for end-processing to generate resected DNA after IR, RAD51 foci staining is usually performed ~10 hours after the IR. As a result, the frequency of IR induced RAD51 foci is also influenced by repair pathway choices (i.e. NHEJ vs homologous recombination), end-resection and cell cycle distributions (since RAD51 filament can only form in S/G2 phase of the cell cycle), in addition to RPA loading. In contrast, HU

directly induces ssDNA, RPA loading and ATR activation, allowing more direct measurement for ATR activity dependent effects.

Taken together, the results from IR vs HU dependent RAD51 foci formation highlights the ssDNA dependent structural function of ATR-KD. We have now included the data in Supplementary Figure 6d and discussed them in page 13.

Reviewer #2 (Remarks to the Author):

I appreciate the authors' response but still feel that the following issues need further attention. The lack of biochemistry is still a problem. The increase of RPA and phospho-RPA on chromatin in ATR+/KD cells is not necessarily a result of reduced RPA exchange - it may be a consequence of increased origin firing and induction of DNA damage. The functional contributions of ATR and RPA exchanges are still difficult to estimate. ATR-KD inhibits phosphorylation of many ATR substrates in addition to affecting ATR and RPA exchanges. It remains possible that the phenotypes of ATR+/KD cells are primarily caused by the reduced phosphorylation of ATR substrates.

We thank the reviewer for his/her efforts to help us consider indirect effect of *Atr*^{+/KD} and emphasize the importance of the biochemical studies. As discussed in the manuscript and below, we acknowledge that the overall phenotypes of the *Atr*^{+/KD} mice, as in all other mouse models, reflect a combination of direct and indirect (including compensatory and 2nd changes) caused by the primary genetic manipulation. In this case, the *Atr*^{+/KD} mouse model reveals tissue specific and developmental stage specific defects that are not found in *Atr*^{+/-} mice and uncovers physiological consequences of Atr-KD. Moreover, the ATR kinase dependent exchange of ATR at DNA damage sites provides a previous unrecognized mechanism to explain this dominant negative structural function of ATR-KD. Specifically, we would like to point out that reduced ATR-KD exchange impairs the activation of ATR kinase pathway and suppresses the exchange of a subset of RPA. Both together, not either alone, contribute to the overall phenotypes, including the enhanced RPA phosphorylation in HU treated *Atr*^{+/KD} cells. We have now included this in the discussion.

We acknowledge that extensive biochemical studies from many experts, likely including reviewer 2, had provided many insights on ATR activation and regulations. The novelty and the emphasis of our current study is the physiological impact (highlighted by the mouse models) and dynamic exchange of ATR (best demonstrated by cell biology studies). Several evidences (listed below) suggest that a biochemical explanation for this “mobility” defect might or might not exist and likely beyond the scope of this already very extensive and comprehensive study.

- 1) Although in all the mutant, ½ of the active kinase is absent, neither *Atr*^{+/-} nor *Chk1*^{+/-} mice have the ssDNA selective spermatogenesis defects or telomere/rDNA instability, strongly suggesting ATR-KD protein physically impairs genomic instability beyond simple reduction of ATR activity and lack of phosphorylation of its main effective kinase.
- 2) The kinase activity dependent exchange of ATR documented here is likely a direct effect of ATR on itself, not mediated by lack of phosphorylation of its other substrates, as ATR inhibitor, but not CHK1 inhibitor, markedly impairs ATR exchange.
- 3) If the “biochemistry” experiments refer to “auto-“ or “trans-“ phosphorylation, the T1989 site previously identified in human ATR is NOT conserved in mouse. Structural analyses of the yeast orthologue of ATR- Mec1 also suggests that this site (corresponding to the T1989) is not near the catalytic center of ATR or its homodimer (Gao et al., 2011) . In related ATM and DNA-PK, the expression of kinase dead protein had markedly different effects than the alanine substitution of the “auto- or trans- phosphorylation sites (Daniel et al., 2012; Daniel et al., 2008; Jiang et al., 2015; Liu et al., 2012; Pellegrini et al., 2006; Zhang et al., 2011). As discussed in the manuscript, structural studies also predict significant conformation changes upon activation, which might regulate exchange independent of “auto- or trans-“ phosphorylation.
- 4) If the biochemistry experiments refer to measure physical interaction of ATR-ATRIP on RPA coated ssDNA, cell biology, especially FRAP experiment, offers similar insights in a physiological setting. We also noted that despite extensive efforts, robust *in vitro* activation of ATR has not yet been achieved. This might be due to the requirement for additional factors – including, but not limited to TopBP1, 9-1-1 complex, ETAA1 or other proteins for the full activation of ATR. Given this technical barrier, we felt the FRAP experiment is better suited than bulk biochemical experiments in measuring mobility.

It is unfortunate that we were not able to guess what feasible and mechanistically critical biochemistry experiments that reviewer 2 is having in mind during both the primary and 2nd review. Hopefully our sincere explanation and all round discussions could cover some of the ground. In summary, we do feel the unique phenotype of the novel *Atr*^{+ /KD} mouse models and the surprising discovery of the kinase dependent exchange of ATR documented in our manuscript provide significant new insights to the activation of ATR and impact of ATR inhibition that would be important for a broad range of readers.

Reviewer #3 (Remarks to the Author):

In this revised and improved version of the manuscript, the authors have addressed most of my requests. However, the authors have failed to show convincing results about proper RAD51 loading on mutant spermatocytes. The images provided in the Figure S2f are uninformative since no clear foci can be observed at 100% (or even at 400%) magnification. Also, based on the proper

achievement of homologous chromosome synapsis displayed by the *Atr*^{+/KD} spermatocytes, I would not expect differences that could be easily seen by the eye. Furthermore, the authors state that “the [...] frequency of RAD51 foci [...] were not significantly altered in the *Atr*^{+/KD} spermatocytes”, but they don’t provide any data about the frequency of RAD51 foci found in controls or mutant spermatocytes. Thus, I think it is crucial to count the number of RAD51 foci found at leptotene stage in wt, hets, and *Atr*^{+/KD} to complete this analysis. Apart from this one, I do not have any other comment on the manuscript, which I am sure it will be of interest to the broad readership of Nature Communications.

We thank the reviewer for the positive comment and for considering our manuscript suitable for publication in Nature Communications. We apologize for limited resolution of the RAD51 staining in the first revision. We have now optimized the staining as well as the image acquisition of the Rad51 staining, quantified the number of RAD51 foci in leptotene stage from *Atr*^{+/+}, *Atr*^{+/-} and *Atr*^{+/KD} mice. Based on the quantification and statistical analyses now in **Supplementary Fig. 2f (also on the right)**, the frequency of RAD51 foci is similar in leptotene stage spermatocytes from *Atr*^{+/+}, *Atr*^{+/-} and *Atr*^{+/KD} mice. This result is consistent with the normal autosome synapsis measured by MLH1 foci (Supplementary Fig. 2e) and normal fertility of the female *Atr*^{+/KD} mice (Supplementary Fig. 1h). Together these and other data supports a X-Y body specific spermatogenesis defects in *Atr*^{+/KD} mice. In addition, we also included representative RAD51 foci staining and quantifications from *Atr*^{+/+} and *Atr*^{+/KD} mice in the revision below.

Change of figure orders -

To provide space in supplementary figure 2 to accommodate the quantification of the RAD51 foci, we moved the ratio of dipotene: pachytene originally in

Supplementary Figure 2d to Figure 1e. Original Figure 1e is now renamed Figure 1f. Original Supplementary Figure 2e and 2g, are now Supplementary Figure 2d and 2e, respectively.

References

- Daniel, J.A., Pellegrini, M., Lee, B.S., Guo, Z., Filsuf, D., Belkina, N.V., You, Z., Paull, T.T., Sleckman, B.P., Feigenbaum, L., *et al.* (2012). Loss of ATM kinase activity leads to embryonic lethality in mice. *J Cell Biol* **198**, 295-304.
- Daniel, J.A., Pellegrini, M., Lee, J.H., Paull, T.T., Feigenbaum, L., and Nussenzweig, A. (2008). Multiple autophosphorylation sites are dispensable for murine ATM activation in vivo. *JCell Biol* **183**, 777-783.
- Gao, D., Inuzuka, H., Tan, M.K., Fukushima, H., Locasale, J.W., Liu, P., Wan, L., Zhai, B., Chin, Y.R., Shaik, S., *et al.* (2011). mTOR drives its own activation via SCF(betaTrCP)-dependent degradation of the mTOR inhibitor DEPTOR. *Mol Cell* **44**, 290-303.
- Jiang, W., Crowe, J.L., Liu, X., Nakajima, S., Wang, Y., Li, C., Lee, B.J., Dubois, R.L., Liu, C., Yu, X., *et al.* (2015). Differential phosphorylation of DNA-PKcs regulates the interplay between end-processing and end-ligation during nonhomologous end-joining. *Mol Cell* **58**, 172-185.
- Liu, X., Jiang, W., Dubois, R.L., Yamamoto, K., Wolner, Z., and Zha, S. (2012). Overlapping functions between XLF repair protein and 53BP1 DNA damage response factor in end joining and lymphocyte development. *Proc Natl Acad Sci U S A* **109**, 3903-3908.
- Pellegrini, M., Celeste, A., Difilippantonio, S., Guo, R., Wang, W., Feigenbaum, L., and Nussenzweig, A. (2006). Autophosphorylation at serine 1987 is dispensable for murine Atm activation in vivo. *Nature*.
- Riballo, E., Kuhne, M., Rief, N., Doherty, A., Smith, G.C., Recio, M.J., Reis, C., Dahm, K., Fricke, A., Krempler, A., *et al.* (2004). A pathway of double-strand break rejoining dependent upon ATM, Artemis, and proteins locating to gamma-H2AX foci. *MolCell* **16**, 715-724.
- Shibata, A., Moiani, D., Arvai, A.S., Perry, J., Harding, S.M., Genois, M.M., Maity, R., van Rossum-Fikkert, S., Kertokallio, A., Romoli, F., *et al.* (2013). DNA double-strand break repair pathway choice is directed by distinct MRE11 nuclease activities. *Mol Cell* **53**, 7-18.
- Zhang, S., Yajima, H., Huynh, H., Zheng, J., Callen, E., Chen, H.T., Wong, N., Bunting, S., Lin, Y.F., Li, M., *et al.* (2011). Congenital bone marrow failure in DNA-PKcs mutant mice associated with deficiencies in DNA repair. *J Cell Biol* **193**, 295-305.

REVIEWERS' COMMENTS:

Reviewer #1 (Remarks to the Author):

This is excellent work and should be published without delay.

Reviewer #3 (Remarks to the Author):

The authors have addressed my suggestions successfully, thus I have no further comments for this version of the manuscript.

Point-by-point response to reviewers.

Original comments from the reviewers are **bold**. Our responses are in *italics*.

Reviewer #1 (Remarks to the Author):

This is excellent work and should be published without delay.

We thank the reviewer for his/her thoughtful suggestions and comments throughout the revision process and for his/her final positive statement.

Reviewer #3 (Remarks to the Author):

The authors have addressed my suggestions successfully, thus I have no further comments for this version of the manuscript.

We thank the reviewer for his/her thoughtful suggestions and comments throughout the revision process and for his/her final positive statement.